# Strassen Attention, Split VC Dimension and Compositionality in Transformers

**Alexander Kozachinskiy**
CENIA
alexander.kozachinskyi@cenia.cl

**Felipe Urrutia**
University of Chile & CENIA
furrutia@dim.uchile.cl

**Hector Jimenez**
University of Chile & CENIA
hjimenez@dcc.uchile.cl

**Tomasz Steifer**
Institute of Fundamental Technological
Research, Polish Academy of Sciences
tsteifer@ippt.pan.pl

**Germán Pizarro**
CENIA
german.pizarro@cenia.cl

**Matías Fuentes**
IMC, Pontifical Catholic University of Chile
mdfuentes4@uc.cl

**Francisco Meza**
IMC, Pontifical Catholic University of Chile
fjmeza1@uc.cl

**Cristian B. Calderon**
CENIA
cristian.buc@cenia.cl

**Cristóbal Rojas**
Institute for Mathematical and Computational Engineering
Pontifical Catholic University of Chile & CENIA
luis.rojas@uc.cl

## Abstract

We propose the first method to show theoretical limitations for one-layer softmax transformers with arbitrarily many precision bits (even infinite). We establish those limitations for three tasks that require advanced reasoning. The first task, Match 3 (Sanford et al., 2023), requires looking at all possible token triplets in an input sequence. The second and third tasks address compositionality-based reasoning: function composition (Peng et al., 2024) and binary relations composition, respectively. We formally prove the inability of one-layer softmax Transformers to solve any of these tasks. To overcome these limitations, we introduce Strassen attention and prove that, equipped with this mechanism, a one-layer transformer can in principle solve all these tasks. Importantly, we show that it enjoys sub-cubic running-time complexity, making it more scalable than similar previously proposed mechanisms, such as higher-order attention (Sanford et al., 2023). To complement our theoretical findings, we experimentally studied Strassen attention and compared it against standard (Vaswani et al, 2017), higher-order attention (Sanford et al., 2023), and triangular attention (Bergen et al. 2021). Our results help to disentangle all these attention mechanisms, highlighting their strengths and limitations. In particular, Strassen attention outperforms standard attention significantly on all the tasks. Altogether, understanding the theoretical limitations can guide research towards scalable attention mechanisms that improve the reasoning abilities of Transformers.

39th Conference on Neural Information Processing Systems (NeurIPS 2025).

# 1  Introduction

**What tasks can Transformers solve?**  A fundamental question in modern AI is understanding why and under which conditions a given deep network architecture succeeds or fails on a given task. Numerous recent works have shown that tasks requiring compositional reasoning stand out as particularly challenging [6, 24, 16, 15]. Compositionality refers to the ability of composing blocks of knowledge to generate new content or solve new tasks, and is believed to play a major role in the emergence of systematic generalization in language [17, 4]; making the question for these kind of tasks even more relevant.

To address this problem, benchmark datasets such as SCAN [14], PCFG [11], CLUTRR [22] or COGS [13] have been introduced. Moreover, different empirical methods have been developed to test, quantify and compare compositional capabilities [12], as well as to assess the impact of design choices on the performance of Transformers on compositional tasks [18]. While many of these works provide strong empirical evidence that Transformers may suffer from inherent limitations for compositional tasks [8], our current theoretical understanding of the underlying phenomenon remains limited.

Here, we take a deeper dive into the problem and aim at contributing to the study of compositionality in Transformers on a more basic mathematical level. As we shall see, this allows us to pin down simple theoretical obstacles that make compositionality hard for the standard Transformer architecture. In turn, we can devise a new attention mechanism, which transcends these basic limitations.

**Related work**  Our work can be related to two complementary lines of research which we now develop.

*Theoretical limitations of Transformers.*  A natural first step to explain why models struggle with some tasks is to study their *expressivity*— whether a given architecture is *in-principle* capable of solving the task in question. That is, whether there exists a choice of parameters that results in a model computing the function underlying the task solutions. If the answer is positive, then the chances are that the difficulty lies in efficiently learning the appropriate parameters. If the answer is negative, then a formal proof of this fact can be directly related to the poor performance observed in practice.

The first theoretical limitations of this sort were obtained for Transformers using *hardmax* attention [10]. Instead of computing attention as a convex combination of all input tokens using *softmax*, one takes a single token where the attention is maximal. Using this simplification, Hahn showed that hardmax Transformers with $O(1)$ layers cannot compute formal languages such as PARITY, MAJORITY, or Dyck-1.

Theoretical limitations against softmax Transformers have recently been obtained by employing a different proof technique, one based on *communication complexity* [19]. The idea is to show that a Transformer solving a given task can be used to construct a corresponding communication protocol for a problem whose communication complexity is known. From this, one obtains lower bounds on the size of Transformers capable of solving the task for inputs of a given length. In [19], for example, the authors apply this technique to show that any one-layer softmax Transformer that can compute the composition of two functions must have $n^{\Omega(1)}$ embedding dimension, where $n$ is the size of the input. Crucially, for this conclusion to hold, one must assume that the Transformer works with a relatively low number of precision bits, namely sub-linear in $n$. The technique has subsequently been applied to show lower bounds for other tasks such as string equality [3] and Match3 [20] (see Section 3.2.2 for a definition).

*Overcoming Transformers limitations.*  Equipped with a better theoretical understanding of why Transformers struggle with some tasks, the next question is how they can guide research towards the construction of more expressive machines. A key observation is that the standard attention mechanism can only see interactions between pairs of tokens, whereas compositional tasks require to reason about the interaction of three or more tokens simultaneously [2, 20]. Consequently, suitably modified attention mechanisms have been proposed, which would track interactions between more than two tokens. For instance, *triangular attention* [2] (see Section 2 for a definition) outperforms standard attention in compositional tasks such as CLUTRR [22] or COGS [13]. Similarly, it has been shown that *higher-order tensor attention* embedded in a constant size one-layer Transformer can theoretically solve Match3, a task that cannot be solved by the standard attention [20]. However,

both these mechanisms suffer from a significant increase in running-time complexity, e.g., order 3 requires cubic-time, limiting its scalability and potentially affecting its practical relevance. In the case of triangular attention, the mechanism works with an adjacency matrix of a graph (in which case it is square-time in the size of the graph). It is possible to produce an adjacency matrix from the sequence of tokens, as already shown in Bergen et al. [2], but this increases the complexity to cubic. The higher-order tensor attention, on top of being cubic-time as well, has been only considered in theoretical work and has not been evaluated empirically yet.

**Our contributions**   In this work we aim at addressing these fundamental questions from complementary angles. Our main results can be summarized as follows:

- **We present a new technique for proving theoretical limitations.** Since previous results establishing limitations for softmax transformers are based on communication complexity, they only apply to transformers that store numbers with relatively few precision bits. This raises the question – would the use of long arithmetic help to circumvent these limitations? In this paper, we answer this question negatively. In order to establish that, we introduce a new technique for proving limitations, based on the novel notion of *Splitting VC dimension*.

  The splitting VC dimension of a Boolean function $f$, denoted by $\mathsf{split\text{-}VC}(f)$, is a positive integer number intended to capture the complexity of $f$ in a certain combinatorial sense. Our main technical contribution is the following:

  ***Main Theorem***. *Let $T$ be any one-layer standard attention Transformer that can do arithmetic operations over real numbers with infinite precision. Denote by $d$, $H$ and $L$, respectively, its embedding dimension, number of attention heads and size of the output MLP of $T$. Suppose $T$ can compute a function $f$. Then it holds that $\max\{d, H, L\} \geq \mathsf{split\text{-}VC}(f)^{\Omega(1)}$.*

  To put it differently, even an idealized transformer $T$ that can manipulate numbers with infinitely many bits cannot compute a function $f$ with large $\mathsf{split\text{-}VC}(f)$ unless $T$ has a large embedding dimension, a large number of attention heads, or a large output MLP. Obviously, the same limitations apply to real-life Transformers that can only manipulate numbers using some finite (however large) number of bits.

- **We obtain new theoretical limitations for tasks requiring complex reasoning**. We apply our new method to the task of function composition (Peng et al. [19]) and the Match3 (Sanford et al. [20]) task, and show that previously known limitations for these tasks are also applicable to transformers that work with arbitrarily large (and even infinite) precision.

  We also do this for a new task, that we name *binary relation composition* task. We introduce it because the function composition task can be solved by a 2-layer transformer, while for the binary relation composition task there is no apparent solution with any constant number of layers. Finally, we introduce another task that we call *Quotient Binary Relation Composition* (see Section 5). Our method allows to establish limitations for this task even for mixed transformers that can use both the standard and the triangular attention.

- **We develop a more scalable higher-order mechanism.** Previous works (e.g. [20]) have shown that higher-order methods can in principle overcome these limitations, but suffer from severe scalability issues. In an attempt to address these difficulties, we present and study *Strassen attention*, a variation devised to be sensitive to interactions between triplets of tokens without sacrificing too much efficiency. In contrast to previous similar attention mechanisms that required running time $n^3$ for inputs of length $n$, our mechanism enjoys $n^2$ space and sub-cubic running time, making it more scalable. At the same time, we show that 1-layer constant-size Strassen attention transformers are theoretically capable of solving all 4 aforementioned tasks.

- **We empirically demonstrate the convergence of Strassen attention.** Is Strassen attention capable of converging to these theoretical solutions during learning? We empirically demonstrate that the answer is positive for all 4 tasks. For comparison, we have additionally evaluated all other attention mechanisms on these tasks. Our empirical findings agree with theory and show that Strassen attention: *(i)* outperforms the accuracy of Standard attention, *(ii)* achieves superior efficiency in terms of training time and computational resources when compared with other triple-wise interaction attention mechanisms.

**Paper organization** In Section 2 we recall the Transformer architecture and introduce our new Strassen attention mechanism. Section 3 presents our new lower bound method and its application to function composition, Match3, and binary relation composition tasks. In Section 4, we show that Strassen attention can be implemented in sub-cubic time, and formally prove that, in principle, it is capable of solving the three aforementioned task. Section 5 presents a novel task allowing to separate the capabilities of Strassen attention from those of standard and triangular attentions. Finally, section 6 contains our experimental findings. Due to space constraints, some proofs and experimental details are deferred to the Appendix.

## 2 Preliminaries

Throughout the paper, we denote $[n] = \{1, \ldots, n\}$ for $n \in \mathbb{N}$. For a set $\Sigma$, we will denote by $\Sigma^n$ the collection of sequences of elements of $\Sigma$ of length $n$, and by $\Sigma^*$ the collection of all finite sequences. We start by briefly recalling the basics of the Transformer architecture and formally defining the attention mechanisms studied in this paper.

The main block of the Transformer layer is the *attention function*, formally defined as a length-preserving function $a \colon (\mathbb{R}^d)^* \to (\mathbb{R}^d)^*$, where $d$ is the embedding dimension. In this paper, we consider 4 types of attention functions.

*Standard attention* [23], receives as input a sequence $x_i \in \mathbb{R}^d, i = 1, \ldots, n$ and outputs a sequence $a_i \in \mathbb{R}^d, i = 1, \ldots, n$, computed as follows:

$$a_i = \sum_{j=1}^{n} a_{ij} v_j \tag{1}$$

$$a_{ij} = \mathsf{Softmax}_j(q_i k_j / \sqrt{d}) \tag{2}$$

$$q_i = W^q x_i, \qquad k_j = W^k x_j, \qquad v_j = W^v x_j, \tag{3}$$

where $W^q, W^k, W^v \in \mathbb{R}^{d \times d}$

*Triangular attention* [2] is defined for $n = m^2$, with input tokens indexed by pairs $(i, j), i, j = 1, \ldots, m$. Given an input $\{x_{ij} \in \mathbb{R}^d\}_{i,j=1}^m$, the output is computed as follows:

$$a_{ij} = \sum_{\ell=1}^{m} a_{i\ell j} v_{i\ell j} \tag{4}$$

$$a_{i\ell j} = \mathsf{Softmax}_\ell(q_{i\ell} k_{\ell j} / \sqrt{d}) \tag{5}$$

$$q_{i\ell} = W^q x_{i\ell}, \qquad k_{\ell j} = W^k x_{\ell j}, \tag{6}$$

$$v_{i\ell j} = V_1 x_{i\ell} \odot V_2 x_{\ell j}, \tag{7}$$

where $W^q, W^k, V_1, V_2 \in \mathbb{R}^{d \times d}$.

*Third-order attention* [20] is computed as follows:

$$a_i = \sum_{j,\ell=1}^{n} a_{ij\ell}(v_j \odot v_\ell) \tag{8}$$

$$a_{ij\ell} = \mathsf{Softmax}_{j,\ell}(q_i(k_j \odot k_\ell) / \sqrt{d}) \tag{9}$$

$$q_i = W^q x_i, \qquad k_j = W_1^k x_j, \qquad k_\ell = W_2^k x_\ell, \tag{10}$$

$$v_j = V_1 x_j, \qquad v_\ell = V_2 x_\ell, \tag{11}$$

where $W^q, W_1^k, W_2^k, V_1, V_2 \in \mathbb{R}^{d \times d}$.

We introduce **Strassen attention**, computed as follows:

$$a_i = \sum_{j,k=1}^{n} a_{ijk}(v_j \odot v_k) \tag{12}$$

$$a_{ijk} = \mathsf{Softmax}_{j,k}((f_i g_j + g_j h_k + h_k f_i)/\sqrt{d}) \tag{13}$$

$$f_i = W^f x_i, \qquad g_j = W^g x_j, \qquad h_k = W^h x_k, \tag{14}$$

$$v_j = V_1 x_j, \qquad v_k = V_2 x_k, \tag{15}$$

where $W^f, W^g, W^h, V_1, V_2 \in \mathbb{R}^{d \times d}$, and $\odot$ denotes the Hadamard product. See Figure 2 in Appendix A for the illustration of these attention mechanisms.

**Definition 2.1.** A one-layer Transformer $T$ with $H$ heads and embedding dimension $d$ is given by $H$ attention functions $Att_1, \ldots, Att_H \colon (\mathbb{R}^d)^* \to (\mathbb{R}^d)^*$, a matrix $W_O \in \mathbb{R}^{d \times (dH)}$, and an "output MLP" $\mathcal{N} \colon \mathbb{R}^d \to \mathbb{R}$ with $P$ parameters, which is formally a neural network with ReLU activation. We define the *size* of $T$ as $size(T) = \max\{H, d, P\}$. The *output* of $T$ on input $\bar{x} = (x_1, \ldots, x_n) \in (\mathbb{R}^d)^n$ is the sequence $\bar{y} = (y_1, \ldots, y_n) \in (\mathbb{R})^n$ given by

$$a_i^{(h)} = (Att_h(\bar{x}))_i, \qquad h = 1, \ldots H \tag{16}$$

$$\widehat{a}_i = W_O \begin{pmatrix} a_i^{(1)} \\ \vdots \\ a_i^{(H)} \end{pmatrix} \tag{17}$$

$$y_i = \mathcal{N}(x_i + \widehat{a}_i). \tag{18}$$

So far, Transformers are defined as functions transforming sequences of vectors in $\mathbb{R}^d$. For the tasks we consider in this paper, we need to apply Transformers on sequences of symbols of an arbitrary finite alphabet $\Sigma$. This is done by including into the Transformer a positional encoding $p \colon [n] \times \Sigma \to \mathbb{R}^d$. For a given $p$, an input word $w = \sigma_1 \ldots \sigma_n \in \Sigma^n$ is converted into a sequence of vectors:

$$x_1 = p(1, \sigma_1), \ldots, x_n = \sigma(n, \sigma_n)$$

that constitute the input for the Transformer. For our lower bounds, we make no assumptions about the function $p$. In our upper bounds, however, we present constructions that use reasonable, easily computable positional encodings of the form $p(i, \sigma_i) = q(i) + r(\sigma_i)$, treating positions and symbols independently.

## 3 Theoretical Limitations of Transformers via Split-VC dimension

We now introduce the notion of splitting dimension for a Boolean function $f$. Let $\mathcal{X}$ be a set and $H \subseteq \{0,1\}^{\mathcal{X}}$ be a collection of functions $h : \mathcal{X} \to \{0,1\}$ which we will refer to as *hypothesis class*. We say that an hypothesis class $H$ *shatters* a subset $X = \{x_1, \ldots, x_m\} \subseteq \mathcal{X}$ if for any Boolean vector $c_1 \ldots c_m \in \{0,1\}^m$ there exists $h \in H$ with $h(x_1) = c_1, \ldots, h(x_m) = c_m$. The maximal $m$ for which $H$ shatters some $X \subseteq \mathcal{X}$ of cardinality $m$ is called the *VC dimension* of $H$ [21].

We now explain how to adapt VC dimension to use it as a complexity measure of a single function (instead of a class of functions). Take a function $f \colon \Sigma^n \to \{0,1\}$ and imagine we split the $n$ arguments of $f$ into two parts. One part will continue to correspond to the inputs, but the other is to be regarded as a set of *parameters*, so that now we can see $f$ as a class of functions that has a well-defined VC dimension. The complexity of $f$ will be defined as the maximal VC dimension that we can obtain in this way, considering all possible splittings into parameters and inputs.

Let us formalize this idea. For a set of positions $A \subseteq \{1, \ldots, n\}$, we define a Boolean matrix $M_f^A$ as follows. Its rows (interpreted as inputs) will be indexed by all the words $w^1 \in \Sigma^A$ and its columns (interpreted as parameters) by the words $w^2 \in \Sigma^B$, where $B = \{1, \ldots, n\} \setminus A$. Thus, $M_A^f$ will be a $|\Sigma|^{|A|} \times |\Sigma|^{n-|A|}$ Boolean matrix. The value of $M_A^f$ at $(w^1, w^2)$ is then defined as

$$M_A^f(w^1, w^2) = f(w^1 \oplus w^2)$$

where $w^1 \oplus w^2 \in \Sigma^n$ is obtained by merging $w^1$ and $w^2$ according to the positions indicated by $A$ and $B$, i.e.

$$(w^1 \oplus w^2)_i = \begin{cases} w_i^1 & i \in A, \\ w_i^2 & i \in B, \end{cases} \qquad i = 1, \dots, n.$$

**Definition 3.1.** We define the *splitting VC dimension* of $f \colon \Sigma^n \to \{0,1\}$, denoted by split-VC$(f)$, as the maximum over $A \subseteq \{1, \dots, n\}$ of the VC dimension of the set of columns of $M_f^A$, understood as Boolean functions on the set of rows.

**Example.** Consider a Boolean function $f \colon \{0,1\}^4 \to \{0,1\}$, defined by $f(x_1, x_2, x_3, x_4) = (x_1 \wedge x_2) \oplus (x_3 \wedge x_4)$. We claim that split-VC$(f) = 2$. To establish split-VC$(f) \geq 2$, we consider $A = \{1, 3\}$. The matrix $M_f^A$ is constructed as follows: its rows are indexed by Boolean vectors $x_1 x_3 \in \{0,1\}^2$, its columns by Boolean vectors $x_2 x_4 \in \{0,1\}^2$, and the intersection of the row $x_1 x_3$ and column $x_2 x_4$ has $f(x_1, x_2, x_3, x_4)$ in it. Thus, this matrix looks as follows:

| $x_1 x_3$ \\ $x_2 x_4$ | 00 | 01 | 10 | 11 |
|---|---|---|---|---|
| 00 | 0 | 0 | 0 | 0 |
| 01 | 0 | 1 | 0 | 1 |
| 10 | 0 | 0 | 1 | 1 |
| 11 | 0 | 1 | 1 | 0 |

Columns of this matrix realize all 4 Boolean functions on the second and the third row, implying that the VC dimension of the set of columns of this matrix is at least 2. We now observe that there is no $A \subseteq \{1, 2, 3, 4\}$ such that the set of columns of $M_f^A$ has VC dimension at least 3. Indeed, this requires $A$ to be of size at least 2, because otherwise there are less than 3 rows. But if $|A| \geq 2$, there are at most 4 columns, making it impossible to realize $2^3 = 8$ different Boolean functions.

## 3.1 Main Theorem

In order to state our main Theorem, we first need to specify a way to see a Transformer as computing a given boolean function $f \colon \Sigma^n \to \{0,1\}$. We assume that an input word $w = \sigma_1 \dots \sigma_n$ is given to the Transformer using $n + 1$ tokens. The first $n$ tokens are used to encode the $n$ symbols of $w$, while the $(n+1)$-st *auxiliary token* (initially encoding the empty symbol) is used to encode the output $f(w)$ of the function being computed. More specifically, the output of the Transformer in the auxiliary token has to be a real number $y_{n+1}$, satisfying $f(w) = \text{sign}(y_{n+1})$. When a Transformer $T$ fulfills this requirement for a given function $f$, we will say that the Transformer $T$ *computes $f$ in an auxiliary token*. We can now state our main result.

**Theorem 3.2.** *Let $T$ be a one-layer standard-attention Transformer and let $f \colon \Sigma^n \to \{0,1\}$ be a Boolean function. If $T$ computes $f$ in an auxiliary token, then $size(T) = $ split-VC$(f)^{\Omega(1)}$.*

*Remark* 3.3. Note that from Theorem 3.2 it follows that for any Transformer $T$ satisfying $size(T) = n^{o(1)}$, it is impossible to compute in an auxiliary token any function $f \colon \Sigma^n \to \{0,1\}$ with split-VC$(f) = n^{\Omega(1)}$.

## 3.2 Applications to three concrete tasks

We now apply Theorem 3.2 to three different tasks.

### 3.2.1 Function Composition

Introduced in Peng et al. [19], in this task we receive a description of two functions $g : [n] \to [n]$ and $h : [n] \to [n]$, and we are asked the value of $h(g(x))$ for a given $x \in [n]$. The task is presented to a Transformer in the form of a sequence of $2n + 1$ tokens, divided in three parts. The first two parts encode the values of $g$ and $h$, and the third part encodes $x$ in a single token, where the output has to be computed. More specifically, the output has to be a real number $y_{2n+1}$ satisfying $|y_{2n+1} - h(g(x))| < 0.5$. That is, with $h(g(x))$ being the closest integer to $y_{2n+1}$.

**Theorem 3.4.** *Let $T$ be any one-layer standard-attention Transformer with $size(T) = n^{o(1)}$. Then $T$ cannot solve the function composition task.*

*Proof.* We show that any Transformer that solves this task can be converted into a Transformer computing in the auxiliary token the following Boolean function:

$$Ind_n \colon [n]^{n+1} \to \{0,1\},$$

$$Ind_n(p_1, q_1, \ldots, q_n) = \begin{cases} 1 & q_{p_1} = 1, \\ 0 & \text{otherwise}, \end{cases}$$

and that this transformation requires adding just $O(1)$ neurons to the output MLP. Indeed, by fixing $x = 1$ and setting $g(1) = p_1, h(1) = q_1, \ldots, h(n) = q_n$, we obtain that the token with $x$ outputs a real number $y$ with $|y - h(g(1))| = |y - q_{p_1}| < 0.5$. It now remains to change the output to $\text{ReLU}(1.5 - y)$ which will be positive exactly when $q_{p_1} = 1$.

The result now follows from Theorem 3.2 and the following:

**Lemma 3.5.** *split-VC*$(Ind_n) \geq n$.

*Proof.* We claim that the VC dimension of the set of columns of $M = M^A_{Ind_n}$ with $A = \{1\}$, is at least $n$. Rows of this matrix are indexed by $p_1 \in [n]$, and columns by vectors $q_1 \ldots q_n \in [n]^n$. We claim that the set of all $n$ rows of $M$ is shattered by the columns of $M$. Take any Boolean vector $b = c_1 \ldots c_n \in \{0,1\}^n$. We need to find $q_1 \ldots, q_n \in [n]^n$ such that $Ind_n(i, q_1, \ldots, q_n) = c_i$ for all $i \in [n]$. It is suffices to define $q_i = \begin{cases} 1 & c_i = 1, \\ 2 & c_i = 0. \end{cases}$ $\square$

$\square$

### 3.2.2 The Match3 task

Next, we define the $\text{Match}_3[n, m]$ task [20]. It is a sequence-to-sequence task. The input is presented to the Transformer as a sequence of $n$ tokens, encoding a vector of integers $(p_1, \ldots, p_n) \in [m-1]^n$. The output is a vector $(y_1, \ldots, y_n) \in \mathbb{R}^n$, required to satisfy:

$$\text{sign}(y_i) = \begin{cases} 1 & \begin{array}{l} \exists j, k \in [n] \text{ s.t.} \\ p_i + p_j + p_k = 0 \pmod{m} \end{array} \\ 0 & \text{otherwise} \end{cases}$$

Note that we deliberately exclude the value $p_i = 0$ for the input positions. This is to avoid inputs that make the task trivial. Indeed, if $p_i = 0$ for some $i$, then the output is trivially 1 since we always have $p_i + p_i + p_i = 0$.

**Theorem 3.6.** *Let $T$ be any one-layer standard-attention Transformer with $size(T) = n^{o(1)}$. Then $T$ cannot solve the $Match_3[n, m]$ task for $m = 2n - 2$.*

### 3.2.3 Binary Relation Composition

Finally, we define the binary relation composition task. This is a sequence-to-sequence task, where on input we get two Boolean matrices $A, B \in \{0,1\}^{\sqrt{n} \times \sqrt{n}}$. The input is presented to a Transformer using $n$ tokens, indexed by pairs $(i, j) \in [\sqrt{n}]^2$, with the $(i, j)$-th token receiving an encoding of $A_{ij}$ and $B_{ij}$. In the output, we have to compute the matrix of the "composition" of $A$ and $B$:

$$B \circ A \in \{0,1\}^{\sqrt{n} \times \sqrt{n}}, \qquad (B \circ A)_{ij} = \bigvee_{k=1}^{\sqrt{n}} (A_{ik} \wedge B_{kj}).$$

More precisely, the output of the $(i, j)$-th token for $(i, j) \in [\sqrt{n}]^2$ has to be a real number $y_{ij}$ with $\text{sign}(y_{ij}) = (B \circ A)_{ij}$. We refer to $A$ and $B$ as "relations" on the set $[\sqrt{n}]$, with $A_{ij}$ indicating whether the pair $(i, j)$ is in the relation $A$ and $B_{ij}$ doing so for relation $B$. For an example, imagine that $A = B$ encodes a relation for a group of researchers where two researchers $i$ and $j$ are related if they have a paper in co-authorship. Then the composition $B \circ A$ refers to the relation of "having a common co-author".

**Theorem 3.7.** *Let $T$ be any one-layer standard-attention Transformer with $size(T) = n^{o(1)}$. Then $T$ cannot solve the binary relation composition task.*

# 4 Strassen attention – An efficient mechanism to solve complex tasks

Both the third-order mechanism of Sanford et al. [20] and the Strassen mechanism define attention as the interaction between three tokens (i.e., a triplet). The crucial difference is that Strassen attention is computed using pairwise dot-products of vectors in the triplet, while the third-order involves coordinates products of all 3 vectors. This allows us to decompose Strassen attention scores in a way that reduces the whole layer to the product of a constant number of $n \times n$ matrices. Famously, the $n \times n$ matrix product admits an $O(n^\omega)$-time algorithm for $w < 3$, with currently best known upper bound on $w$ being 2.372 [7].

**Theorem 4.1.** *Strassen attention layer can be implemented in $O(n^\omega \cdot d)$-time, where $n$ is the number of input tokens, $d$ is the embedding dimension, and $\omega$ is the matrix multiplication exponent, i.e, the smallest real number such that the $n \times n$ matrix product admits an $O(n^\omega)$-time algorithm.*

*Proof.* Writing $a_i$ in (12–15) by definition, we get:

$$a_i = \frac{\sum\limits_{j,k} e^{(f_i g_j + g_j h_k + h_k f_i)/\sqrt{d}}(v_j \odot v_k)}{\sum\limits_{j,k} e^{(f_i g_j + g_j h_k + h_k f_i)/\sqrt{d}}}. \tag{19}$$

Defining matrices $X, Y, Z \in \mathbb{R}^{n \times n}$ and $\widehat{Y} \in \mathbb{R}^{n \times n \times d}$ by:

$$X_{ij} = e^{f_i g_j/\sqrt{d}}, \ Y_{j,k} = e^{g_j h_k/\sqrt{d}}, \ Z_{k,i} = e^{h_k f_i/\sqrt{d}},$$

$$\widehat{Y}_{j,k} = e^{g_j h_k/\sqrt{d}} v_j \odot v_k,$$

we get an expression for $a_i$ in terms of their matrix products $a_i = \frac{(X\widehat{Y}Z)_{ii}}{(XYZ)_{ii}}$. $\qquad\square$

On top of exhibiting a faster running-time, we now show that, in contrast to standard attention, Strassen attention allows a one-layer Transformer to solve all the 3 tasks described in Section 3.2.

**Theorem 4.2.** *The function composition, the binary relation composition, and the $Match_3[n, poly(n)]$ tasks can be solved by a one-layer constant-size Strassen attention Transformer.*

# 5 Disentangling Strassen from Standard and Triangular attentions

So far, we have evaluated tasks that are challenging for one-layer standard attention Transformers but (in principle) easy for one-layer Strassen attention Transformers. In this section, we extend our analysis to triangular attention. As a reminder, running the triangular attention mechanism on a general sequence of length $n$, requires the creation of $n^2$ tokens. In this regime, the triangular attention running time becomes $n^3$. However, when the input is already structured as a $\sqrt{n} \times \sqrt{n}$ matrix, one can run the triangular attention on it directly, making the running time $O(n^{3/2})$. One such task example using structured input is the binary relation composition task. In this case, a one-layer triangular attention can perform this task with one attention head, constant embedding dimension and constant-size output MLP.

We devise a variant of the binary relation task that allows us to disentangle the performance of Strassen attention with that of the triangular and standard attentions, namely the *quotient binary relation composition task*. The latter takes as inputs two Boolean matrices $A, B \in \{0,1\}^{m \times m}$ and a "coloring" function $\mathsf{col}: [m] \to [m]$, where $m = \sqrt{n}$. There are $n = m^2$ input tokens, indexed by pairs from $[m]^2$, with the $(i, j)$-th token receiving $A_{i,j}, B_{i,j}$ and $(\mathsf{col}(i), \mathsf{col}(j))$ as inputs. The quotient of the composition $B \circ A$ by $\mathsf{col}$ is a Boolean matrix $B \circ A/\mathsf{col} \in \{0,1\}^{m \times m}$, defined by:

$$(B \circ A/\mathsf{col})_{ij} = \begin{cases} 1 & \exists k_1, k_2 \in [m] \text{ s.t. } A_{ik_1} = B_{k_2 j} = 1, \\ & \mathsf{col}(k_1) = \mathsf{col}(k_2), \text{ and } k_1 \neq k_2, \\ 0 & \text{otherwise.} \end{cases}$$

The task is to output, for all $(i, j) \in [m]^2$, a real number $y_{ij}$ such that $(B \circ A/\mathsf{col})_{ij} = \mathsf{sign}(y_{ij})$.

To illustrate an instance of this task, imagine that $A = B$ encodes the graph of co-authorship between a set of researchers, and $c$ assigns each researcher its university. Then we have $(B \circ A/\mathsf{col})_{ij} = 1$ if

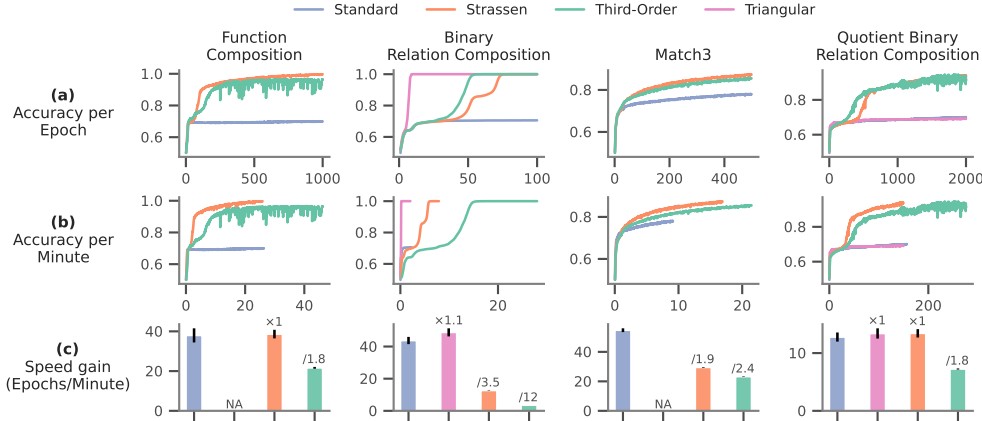

Figure 1: Accuracy as a function of (a) the number of epochs, (b) training time (forward plus backward runtime per epoch), and (c) speed gain (measured as epochs per minute). Performance for each task is presented as the median accuracy over 8 independent runs on data outside the training set. For readability, we truncated the binary relation composition and quotient binary relation composition learning curves to 100 and 2000 epochs, respectively.

and only if researchers $i$ and $j$ have co-authors from the same university, with the condition that these co-authors must be different people.

**Theorem 5.1.** *The quotient binary relation composition task is solvable by a one-layer Strassen-attention constant-size transformer. At the same time, this task cannot be solved by any one-layer Transformer $T$ with $n^{o(1)}$ standard-attention heads, $n^{o(1)}$ triangular-attention heads, $n^{o(1)}$ embedding dimension, and $n^{o(1)}$-size output MLP.*

## 6 Experiments and Results

In this section, we systematically compare the performance of standard, triangular, third-order and Strassen attention in four tasks: (i) Indicator of the 1st coordinate in the Function Composition (with $f = g$ ), (ii) Binary Relation Composition (with $A = B$), (iii) Match3 (over position-aware permutations) and (iv) Quotient Binary Relation Composition (with $B = A^T$). To obtain a tighter upper bound on the capability of the standard attention, we chose to implement simple special cases of the tasks (see Appendix C for all the details on data generation). To compare these attention mechanisms, we evaluate their performance (accuracy per epoch) and training time (accuracy per minute) on the test sets of these tasks. Furthermore, Appendix D provides a thorough analysis in terms of computational performance, evaluating forward pass times, GPU and memory usage. Code for our experiments can be found at ⍟ furrutiav/strassen-attention-neurips25.

Figure 1 displays our main experimental results. First, both the Strassen and third-order attentions are the only two mechanisms that present high accuracy levels across all tasks. Note that Strassen attention displays slight advantages on the Function Composition and Match3 tasks (Figure 1a). Second, Strassen attention consistently outperforms third-order attention in training time (to peak performance), across all tasks (Figure 1b). Third, the advantages of Strassen attention are further exemplified by displaying speed gains (i.e., number of epochs per minute) that match that of the standard attention in the Function Composition and Quotient Binary Relation Composition tasks, and are always superior to that of the third-order attention (Figure 1c). Fourth, perhaps unsurprisingly, triangular attention outperforms all attention mechanisms in terms of training time at the Binary Relation Composition task (Figure 1b, second column). Indeed, Strassen and the third-order attentions need to learn the triangular structure of the task, a knowledge that is already structurally embedded in the triangular attention. Fifth, although the third-order attention presents similar accuracy per epoch trend to that of Strassen attention, its learning dynamics seems to be significantly more unstable, particularly for the Function Composition and Quotient Binary Relation Composition tasks. Sixth, a clear advantage of Strassen attention over the triangular and standard attentions was observed for the Quotient Binary Relation Composition task (Figure 1b, last column). Lastly, it is noteworthy that the

triangular attention framework has a smaller applicability scope, and therefore could not be run on the Function Composition and Match3 tasks (without changing the data presentation from sequences to matrices).

**Strassen vs. standard attention Parameter-matched comparisons**    Given that 1-layer Strassen and standard attention architectures are not parameter matched, we performed two additional experiments to provide a fair comparison between them. In both experiments, Quotient Binary Relation Composition and Binary Relation Composition, we pitted a 1-layer Strassen attention against a 2-layer standard attention, both models with hidden dimension 16. Table 1 revealed that even with less parameters, 1-layer Strassen attention continues outperforming a 2-layer Standard attention (with about 40% more parameters) in both of the tasks considered. Furthermore, to demonstrate the value of Strassen attention over standard attention in more practical settings, we evaluated this two attention architectures on the COGS dataset, in a parameter-matched configuration. Table 1 shows that Strassen attention outperforms standard attention.

| Task | Attention Type | Layers | Hidden dim. | Parameters | Accuracy (%) |
|------|----------------|--------|-------------|------------|--------------|
| Binary Relation | Strassen | 1 | 16 | 2.5k | 100 |
| Composition | Standard | 2 | 16 | 3.6k | 92 |
| Quotient Binary | Strassen | 1 | 16 | 2.5k | 98 |
| Relation Composition | Standard | 2 | 16 | 3.6k | 82 |
| COGS | Strassen | 3 | 64 | 99k | 72 |
| | Standard | 3 | 68 | 99k | 65 |

Table 1: Accuracies for the Binary Relation Composition task, Quotient Binary Relation Composition task and COGS dataset as a function of attention type, layer number, hidden dimension and parameters number.

**Future directions and limitations**    Our results pave the way for several directions of future research. First, we have introduced a new method to obtain limitations for one-layer Transformer, based on a new concept we have named splitting VC dimension. We expect our method will be applied to obtain lower bounds in other architectures and other tasks involving complex reasoning. Second, given the differences observed in learning stability between the third-order and Strassen attentions, the latter seems to be associated with a smoother loss landscape, an hypothesis that needs to be confirmed and studied. Third, Strassen attention can be adapted to incorporate interactions that involve more than three tokens, possibly capturing more complex patterns in data. Yet, practical learnability of such higher-order interactions needs to be assessed. Finally, and related to the previous point, although the main goal of our work was to gain a deeper theoretical understanding of the abilities of the Transformer, our conclusions are limited by using toy tasks. Our next step is to test the theoretical advantages of the Strassen attention using specific benchmarks or even in learning methods such as masked language modeling. At the same time, the possible applications of the Strassen attention are not limited to compositionality but could extend to such areas as knowledge graphs, protein structure prediction and others.

**Acknowledgments**    Kozachinskiy, Urrutia, Jimenez, Pizarro, Calderon, and Rojas are funded by the National Center for Artificial Intelligence CENIA FB210017, Basal ANID. Kozachinskiy is supported by ANID Fondecyt Iniciación grant 11250060.

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

# A Illustration of the attention mechanisms

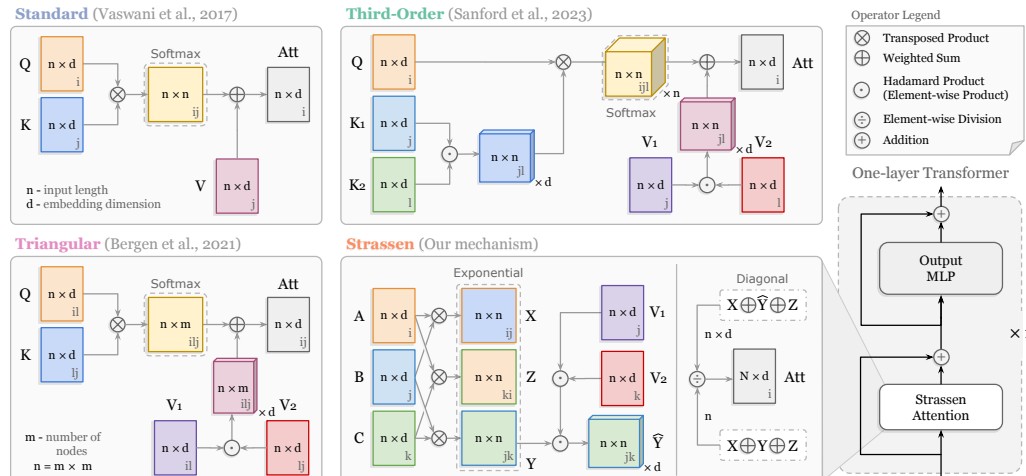

Figure 2: Comparison of Transformer attention mechanisms: **Standard** [23], **Third-Order** [20], **Triangular** [2], and **Strassen** (proposed). The diagram shows operations and the one-layer Transformer architecture, highlighting the flow through each attention and output MLP (without layer normalization and dropout in the attention mechanism). Notation: input length $n$, embedding dimension $d$, number of nodes $m$, with $n = m^2$ when $n$ refers to the shape of flatten matrices.

# B Missing proofs

## B.1 Proof of Theorem 3.2.

Denote $m = \mathsf{split\text{-}VC}(f)$. Let $A \subseteq \{1, \ldots, n\}$ be such that the VC dimension of the set of columns of $M_f^A$ is $m$. Denote $B = [n] \setminus A$. Assume for contradiction that there exists a one-layer standard-attention Transformer $T$ that computes $f$ and whose size (that is, embedding dimension, number of attention heads, and the size of the output MLP) is $m^{o(1)}$.

Consider any $w^1 \in \Sigma^A, w^2 \in \Sigma^B$ and define $w = w^1 \oplus w^2 = \sigma_1 \ldots \sigma_n \in \Sigma^n$. For $h = 1, \ldots, H$, observe that the output of the $h$-th attention head in the $(n+1)$-st token (the auxiliary one, where the output of the function is computed), can be written as:

$$a_{n+1}^{(h)} = \frac{\alpha^{(h)}(w^1) + \beta^{(h)}(w^2) + \gamma^{(h)}}{\lambda^{(h)}(w^1) + \mu^{(h)}(w^2) + \nu^{(h)}}, \tag{20}$$

where

$$\alpha^{(h)}(w^1) = \sum_{i \in A} e^{q_{n+1} k_i} v_i \in \mathbb{R}^d, \qquad \beta^{(h)}(w^2) = \sum_{i \in B} e^{q_{n+1} k_i} v_i \in \mathbb{R}^d$$

$$\lambda^{(h)}(w^1) = \sum_{i \in A} e^{q_{n+1} k_i} \in \mathbb{R}, \qquad \mu^{(h)}(w^2) = \sum_{i \in B} e^{q_{n+1} k_i} \in \mathbb{R}$$

$$\gamma^{(h)} = e^{q_{n+1} k_{n+1}} v_{n+1}, \qquad \nu^{(h)} = e^{q_{n+1} k_{n+1}}$$

Note that that $k_i = W^q p(i, \sigma_i), v_i = W^v p(i, \sigma_i)$ are functions of $w^1$ for $i \in A$ and of $w^2$ for $i \in B$, whereas $q_{n+1}, k_{n+1}$, and $v_{n+1}$ are fixed.

The output of the function is thus computed by:

$$f(w) = M_f^A(w^1, w^2) = \mathsf{sign}\left(\mathcal{N}\left(x_{n+1} + W_O \begin{pmatrix} \frac{\alpha^{(1)}(w^1) + \beta^{(1)}(w^2) + \gamma^{(1)}}{\lambda^{(1)}(w^1) + \mu^{(1)}(w^2) + \nu^{(1)}} \\ \vdots \\ \frac{\alpha^{(H)}(w^1) + \beta^{(H)}(w^2) + \gamma^{(H)}}{\lambda^{(H)}(w^1) + \mu^{(H)}(w^2) + \nu^{(H)}} \end{pmatrix}\right)\right), \tag{21}$$

where $x_{n+1} = p(\varnothing)$, $W_O \in \mathbb{R}^{d \times dH}$ are fixed, and $\mathcal{N}$ is the output MLP of $T$.

Consider now arbitrary real vectors

$$\alpha = (\alpha^{(1)}, \ldots, \alpha^{(H)}) \in \mathbb{R}^{dH}, \qquad \beta = (\beta^{(1)}, \ldots, \beta^{(H)}) \in \mathbb{R}^{dH}$$
$$\lambda = (\lambda^{(1)}, \ldots, \lambda^{(H)}) \in \mathbb{R}^{H}, \qquad \mu = (\mu^{(1)}, \ldots, \mu^{(H)}) \in \mathbb{R}^{H}$$

and define a function $F \colon (\alpha, \lambda, \beta, \mu) \mapsto \{0, 1\}$ as in (21), but with vectors $\alpha, \lambda, \beta, \mu$ allowed to take arbitrary values:

$$F(\alpha, \lambda, \beta, \mu) = \mathsf{sign}\left(\mathcal{N}\left(x_{n+1} + W_O \begin{pmatrix} \frac{\alpha^{(1)} + \beta^{(1)} + \gamma^{(1)}}{\lambda^{(1)} + \mu^{(1)} + \nu^{(1)}} \\ \vdots \\ \frac{\alpha^{(H)} + \beta^{(H)} + \gamma^{(H)}}{\lambda^{(H)} + \mu^{(H)} + \nu^{(H)}} \end{pmatrix}\right)\right), \tag{22}$$

Let $H$ be a class, defined by (22) when $\alpha, \lambda$ are considered as inputs to hypotheses and $\beta, \mu$ as parameters:

$$H = \{h_{\beta,\mu} \colon \mathbb{R}^{dH+H} \to \{0,1\} : h_{\beta,\mu}(\alpha, \lambda) = F(\alpha, \lambda, \beta, \mu), \ (\beta, \mu) \in \mathbb{R}^{dH+H}\}.$$

On the one hand, the VC dimension of $H$ is at least $m = \mathsf{split\text{-}VC}(f)$. Indeed, consider $H$ is an infinite matrix, with rows indexed by $(\alpha, \lambda) \in \mathbb{R}^{dH+H}$, columns by $(\beta, \mu) \in \mathbb{R}^{dH+H}$, and the intersection of the $(\alpha, \lambda)$-row and $(\beta, \mu)$-column containing $F(\alpha, \lambda, \beta, \mu) = h_{\beta,\mu}(\alpha, \lambda)$. The VC dimension of $H$ is the VC dimension of the columns of this matrix. Since by (21) this matrix has $M_f^A$ as a sub-matrix, we get the required lower bound.

On the other hand, the VC dimension of $H$ can be upper bounded by $m^{o(1)}$. Indeed, the number of parameters of $H$ is $dH + H = m^{o(1)}$. To compute the value of a given hypothesis on a given input, it is enough to do $m^{o(1)}$ basic arithmetic operations and comparisons with 0, because the size of $\mathcal{N}$ is $m^{o(1)}$. By Theorem 2.3 in [9], the VC dimension is polynomial in these quantities, which gives us $m^{o(1)}$ upper bound in our case.

## B.2 Proof of Theorem 3.6

We fix the value of $p_1$ to be equal to 1, and consider the output of $\mathrm{Match}_3[n, m]$ at the first position. If we additionally set $p_j \neq m - 2$ for $j \geq 2$, we obtain $p_1 + p_j + p_k = 0 \pmod{m}$ if and only if $j, k \geq 2$ and $p_j + p_k = -1 \pmod{m}$. Hence, a Transformer for the $\mathrm{Match}_3[n, m]$ task can be converted into a Transformer, computing the following function in the auxiliary token:

$$\mathrm{Sum}_2[\ell, m] \colon ([m-1] \setminus \{m-2\})^\ell \to \{0, 1\}$$
$$\mathrm{Sum}_2[\ell, m](p) := \begin{cases} 1 & \exists j, k \in [\ell] \text{ s.t. } p_j + p_k = -1 \pmod{m}, \\ 0 & \text{otherwise} \end{cases}$$

where $\ell = n - 1$.

The following Lemma establishes what we need in order to obtain the desired lower bound from Theorem 3.2.

**Lemma B.1.** *For even $\ell$, it holds that $\mathsf{split\text{-}VC}(\mathrm{Sum}_2[\ell, 2\ell]) \geq \ell/2$.*

*Proof.* We claim that the VC dimension of the set of columns of $M = M_{\mathrm{Sum}_2[\ell, 2\ell]}^A$ with $A = \{1, \ldots, \ell/2\}$, is at least $\ell/2$. The rows of this matrix are indexed by vectors $p = p_1 \ldots p_{\ell/2} \in ([2\ell - 1] \setminus \{2\ell - 2\})^{\ell/2}$, and columns by vectors $q = q_{\frac{\ell}{2}+1} \ldots q_\ell \in ([2\ell - 1] \setminus \{2\ell - 2\})^{\ell/2}$. Note that in this case, for a given row $p$ and column $q$, their merging $p \oplus q$ is simply their concatenation.

We now consider $\ell/2$ rows, corresponding to vectors:

$$p^1 = (2, 1, 1, \ldots, 1),$$
$$p^2 = (1, 4, 1, \ldots, 1),$$
$$\vdots$$
$$p^{\ell/2} = (1, 1, 1, \ldots, \ell).$$

and show that these rows can be shattered by the columns of $M^A_{Sum_2[\ell,2\ell]}$. For any Boolean vector $c_1 \ldots c_{\ell/2} \in \{0,1\}^{\ell/2}$, we have to find a column $q \in ([2\ell-1] \setminus \{2\ell-2\})^{\ell/2}$ such that:

$$Sum_2[\ell, 2\ell](p^i \oplus q) = c_i$$

for all $i = 1, \ldots, \ell/2$. It then suffices to choose $q$ such that

$$q_{\frac{\ell}{2}+i} = \begin{cases} 2\ell - 2i - 1 & c_i = 1, \\ 1 & \text{otherwise.} \end{cases} \qquad i \in [\ell/2].$$

Indeed, first note that the value $m - 2 = \ell - 2$ is not used. Now, if $c_i = 1$, then $p_i \oplus q$ has numbers $2i$ and $2\ell - 2i - 1$, summing up to $2\ell - 1$. Next, if $c_i = 0$, in $p_i \oplus q$ only two numbers can appear, 1 and $2i\ell$, whose sum is neither $2\ell - 1$ nor $4\ell - 1$ because $i \leq \ell/2$. This completes the proof. □

## B.3 Proof of Theorem 3.7

We consider a subproblem of this task, where only elements $A_{12}, \ldots, A_{1k}$ and $B_{21}, \ldots, B_{k1}$ can be equal to 1, where $k = \sqrt{n}$. Under this restriction, a Transformer solving the binary relation composition computes, in the token at position $(1,1)$, the function $\mathrm{Disj}_m \colon \{0,1\}^m \times \{0,1\}^m \to \{0,1\}$ given by

$$\mathrm{Disj}_m(a,b) = \bigvee_{k=1}^{m} (a_i \wedge b_i),$$

with $a$ and $b$ being written in positions $A_{12}, \ldots, A_{1k}$ and $B_{21}, \ldots, B_{k1}$, respectively. In our case, $m = k - 1 = \sqrt{n} - 1$. The results now follows from Theorem 3.2 and the following lemma.

**Lemma B.2.** *split-VC*$(\mathrm{Disj}_m) \geq m$.

*Proof.* We show that the VC dimension of the set of columns of $M^A_{\mathrm{Disj}_m}$ is at least $m$ for $A = \{1, \ldots, m\}$. Both the rows and the columns of $M^A_{\mathrm{Disj}_m}$ are indexed by $m$-bit vectors. We show that $m$ rows, corresponding to the following vectors:

$$a^1 = (1, 0, \ldots, 0, 0),$$
$$a^2 = (0, 1, \ldots, 0, 0),$$
$$\vdots$$
$$a^m = (0, 0, \ldots, 0, 1)$$

can be shattered by the columns of the matrix. To establish that, for every $c \in \{0,1\}^m$ we have to provide $b = b_1 \ldots b_m \in \{0,1\}^m$ with

$$\mathrm{Disj}_m(a^i, b) = c_i, \qquad i \in [m].$$

This can be achieved by simply setting $b_i = c_i$ for $i \in [m]$. □

## B.4 Proof of Theorem 4.2

**Function composition** In the function composition task, we get a $(2n+1)$-length sequence of numbers

$$\phi(1), \ldots, \phi(2n+1) \in [n].$$

The task is to output, in the $(2n+1)$-st token, the value of $h(g(x))$ with the 0.5-precision (in fact, we will do this with a much better precision, namely $e^{-\Omega(n^2)}$), where $g, h \colon [n] \to [n]$ and $x \in [n]$ are such that $g(1) = \phi(1), \ldots, g(n) = \phi(n), h(1) = \phi(n+1), \ldots, h(n) = \phi(2n), x = \phi(2n+1)$.

We use the following positional encoding:

$$x_i = \begin{pmatrix} i \\ i^2 \\ \phi(i) \\ (\phi(i))^2 \\ 1 \\ 0 \\ 0 \end{pmatrix}, \qquad i = 1, \ldots, 2n+1.$$

We take matrices $W^f, W^g, W^h$ in (12–15) so that:

$$f_i = n \begin{pmatrix} (\phi(i))^2 \\ 2\phi(i) \\ -1 \\ 0 \\ 0 \\ 0 \end{pmatrix}, \qquad g_j = n \begin{pmatrix} -1 \\ j \\ j^2 \\ (\phi(j))^2 \\ 2\phi(j) \\ -1 \end{pmatrix},$$

$$h_k = n \begin{pmatrix} 0 \\ 0 \\ 0 \\ -1 \\ k - n \\ k^2 - 2kn + n^2 \end{pmatrix}$$

We obtain:

$$a_{i,j,k} = \mathsf{Softmax}_{j,k} \frac{f_i g_j + g_j h_k + h_k f_i}{\sqrt{6}}$$

$$= \mathsf{Softmax}_{j,k} \frac{-n^2 \left[ (\phi(i) - j)^2 - (\phi(j) - (k - n))^2 \right]}{\sqrt{6}}$$

In particular, the maximum of $a_{2n+1,j,k}$ is for $j$ and $k$ such that $j = \phi(2n + 1) = x$, and $k = n + \phi(j) = n + \phi(x) = n + g(x)$, and other values of $a_{2n+1,j,k}$ are by an $e^{\Omega(n^2)}$-factor smaller. Hence, with precision $\pm e^{-\Omega(n^2)}$, we obtain $a_{2n+1} = v_j \odot v_k$ for $j = x$ and $k = n + g(x)$. Observe that $\phi(k) = \phi(n + g(x)) = h(g(x))$, so it is enough for $v_j \odot v_k$ to have a coordinate equal to $\phi(k)$. We can achieve this by setting matrices $V_1, V_2$ in (15) such that the first coordinates of $v_j$ and $v_k$ are 1 and $\phi(k)$, respectively.

**Binary relation composition**    In this task, the length of input is a square number $n = m^2$, and tokens are indexed by pairs $(i, j) \in [m]^2$. The token, indexed by $(i, j)$, receives on input two bits $A_{ij}, B_{ij}$ from two Boolean matrices $A, B \in \{0, 1\}^{m \times m}$. As an output, the $(i, j)$-th token has to produce a real number $y_{ij}$ such that

$$(B \circ A)_{ij} = \bigvee_{k=1}^{m} (A_{ik} \wedge B_{kj}) = \mathsf{sign}(y_{ij}).$$

We employ the following positional encoding:

$$x_{ij} = \begin{pmatrix} A_{ij} \\ B_{ij} \\ i \\ i^2 \\ j \\ j^2 \\ 1 \end{pmatrix}, \qquad i, j = 1, \ldots, m.$$

We then take matrices $W^f, W^g, W^h$ in (12–15) so that:

$$f_{ij} = n^2 \begin{pmatrix} i^2 \\ 2i \\ -1 \\ \hline j^2 \\ 2j \\ -1 \\ \hline 0 \\ 0 \\ 0 \\ 0 \\ \hline 1/m^2 \\ 1/m^3 \\ 1/m^4 \\ 1/m^5 \end{pmatrix}, \qquad g_{cd} = n^2 \begin{pmatrix} -1 \\ c \\ c^2 \\ \hline 0 \\ 0 \\ 0 \\ \hline d^2 \\ 2d \\ -1 \\ 1 \\ \hline A_{cd} \\ c \\ d \\ 0 \\ 0 \end{pmatrix}, \qquad h_{k\ell} = n^2 \begin{pmatrix} 0 \\ 0 \\ 0 \\ \hline -1 \\ \ell \\ \ell^2 \\ \hline -1 \\ k \\ k^2 \\ \hline B_{k\ell} \\ 1 \\ 0 \\ 0 \\ k \\ \ell \end{pmatrix}$$

(horizontal lines are added for readability). We obtain:

$$f_{ij}g_{cd}+g_{cd}h_{k\ell}+h_{k\ell}f_{ij} = n^4 \left[ -(i-c)^2 - (d-k)^2 - (\ell-j)^2 + A_{cd} + B_{k\ell} + \frac{cm^3 + dm^2 + km + \ell}{m^5} \right].$$
(23)

The expression in brackets has the "integral part", and also has the term $\frac{cm^3+dm^2+km+\ell}{m^5}$ which is $O(1/m)$ and is different for different quadruples $(c, d, k, \ell)$. Hence, there is a unique quadruple $(c, d, k, \ell)$, establishing the maximum of the above expression, and it also maximizes the "integral part" $\left[ -(i-c)^2 - (d-k)^2 - (\ell-j)^2 + A_{cd} + B_{k\ell} \right]$. Because of the factor $n^4$, the value for the other quadruples will be smaller by at least $\Omega(n^4/m^5) = \Omega(n^{3/2})$.

The quantity $\left[ -(i-c)^2 - (d-k)^2 - (\ell-j)^2 + A_{cd} + B_{k\ell} \right]$ is at most 2, being equal to 2 if and only if $c = i, \ell = j, d = k$ and $A_{id} = B_{dj} = 1$. In other words, the maximum of the integral part is 2 if and only if $(B \circ A)_{ij} = 1$.

As a result, the $(i, j)$-th token will get the value of the Hadamard product $v_{cd} \odot v_{k\ell}$ with precision $e^{-\Omega(n^{3/2})}$ for some quadruple $c, d, k, \ell \in [m]$ satisfying:

$$(c = i, \ell = j, d = k \text{ and } A_{cd} = B_{k\ell} = 1) \iff (B \circ A)_{ij} = 1 \tag{24}$$

It remains to define matrices $V^1, V^2$ so that this Hadamard product in some coordinates has $c, d, k, \ell, A_{cd}, B_{k\ell}$, and has 0 where $x_{ij}$ has $i$ and $j$. Then $x_{ij} + (v_j \odot v_k)$ will have all quantities involved in the equalities of the left-hand side of (24), and checking them can be done with a constant-size MLP.

**Match3** On input, we get an array $p_1 \ldots p_n \in [m-1]^n$. We first describe how to check, for a fixed $\Sigma$ and for every $i = 1, \ldots, n$, if there exist $j, k \in [n]$ such that $p_i + p_j + p_k = \Sigma$ using one Strassen attention head. We get a solution for the Match$_3[n, m]$ task with 2 attention heads by applying this construction to $\Sigma = m$ and $\Sigma = 2m$.

The embedding dimension will be 8. Define $q_i = p_i - \Sigma/3$. We use the following positional encoding:

$$x_i = \begin{pmatrix} i \\ q_i \\ q_i^2 \\ 1 \\ 0 \\ 0 \\ 0 \\ 0 \end{pmatrix}, \qquad i = 1, \ldots, n.$$

We define matrices $W^f, W^g, W^h$ in (12–15) so that:

$$f_i = n^2 \begin{pmatrix} -q_i \\ -q_i \\ 0 \\ -q_i^2 \\ 0 \\ 0 \\ 0 \\ 0 \end{pmatrix}, \ g_j = n^2 \begin{pmatrix} 2q_j \\ 0 \\ -q_j \\ 1 \\ -q_j^2 \\ 1 \\ \frac{i}{n^2} \\ 1 \end{pmatrix}, \ h_k = n^2 \begin{pmatrix} 0 \\ 2q_k \\ 2q_k \\ 0 \\ 1 \\ -q_k^2 \\ 1 \\ \frac{k}{n^3} \end{pmatrix}.$$

As a result, we get:

$$a_{ijk} = \mathsf{Softmax}_{j,k} \frac{f_i g_j + g_j h_k + h_j f_i}{\sqrt{8}}$$

$$= \mathsf{Softmax}_{j,k} \frac{n^4 \left( -(q_i + q_j + q_k)^2 + \frac{in+j}{n^3} \right)}{\sqrt{8}}$$

$$= \mathsf{Softmax}_{j,k} \frac{n^4 \left( -(p_i + p_j + p_k - \Sigma)^2 + \frac{in+j}{n^3} \right)}{\sqrt{8}}$$

For a given $i$, the maximum of $a_{ijk}$ is attained on a single triple $(i, j, k)$ with the minimal value of $|p_i + p_j + p_k - \Sigma|$ across the array, and it will be by an $e^{\Omega(n)}$-factor larger than any other value of $a_{i,j,k}$. We added the fraction $\frac{in+j}{n^3}$ to ensure uniqueness of the maximum; the added term is different for different pairs $(i, j)$ while not exceeding $O(1/n)$.

Since all numbers under consideration are polynomial in $n$, the output $a_i$ will be equal to $v_j \odot v_k$ for the maximal pair $(j, k)$ up to $\exp\{-\Omega(n)\}$-precision. In the output MLP, we have to check if $p_i + p_j + p_k = \Sigma$ for this pair $(j, k)$. It is enough to define $V_1$, $V_2$ so that the 5th and the 6th coordinate of $v_j$ and $v_k$ are $1, p_j$ and $p_k, 1$, respectively. As a result, the 2nd, the 5h, and the 6th coordinates of $x_i + a_i$ will be $-p_i, p_k$, and $p_j$, respectively, allowing us to find out if $p_i + p_j + p_k = \Sigma$ with a constant-size output MLP.

### B.5  Proof of Theorem 5.1

**Upper bound**  The upper bound is very similar to our construction for the binary relation composition task in Theorem 4.2. Namely, first we replace $-(d - k)^2$ by $-(\mathsf{col}(d) - \mathsf{col}(k))^2$ in (23). After this modification, the maximum of (23) will be attained on a single quadruple $(c, d, k, \ell)$, and for this quadruple we will have $c = i, \ell = j, \mathsf{col}(d) = \mathsf{col}(k)$ and $A_{id} = B_{kj} = 1$ if and only if a quadruple, satisfying these equalities, exists. However, due to the definition of our task, we have to add a smaller term, enforcing that among quadruples, satisfying this property, those with $d \neq k$ have a larger value of (23). This can be achieved by adding a term of the form $(d - k)^2/m^3$, which is always $O(1/m)$ so that the largest possible difference in this term is smaller than the smallest possible difference of the "integral part" in (23).

We also have to add a term which ensures that the maxima is attained at the unique quadruple. The largest possible difference in this term should be smaller than the smallest possible difference in the previous terms, which is $\Omega(1/m^3)$. We can again take the expression $cm^3 + dm^2 + km + \ell$ but divided by a larger denominator, for instance:

$$\frac{cm^3 + dm^2 + km + \ell}{m^8}$$

(now the maximal possible difference in this term is $O(1/m^4)$). Finally, it remains to multiply all the coefficients by a sufficiently large factor to make the minimal possible difference between the maximum and the other values polynomial in $n$.

**Lower bound**  We employ the same technique as in Theorem 3.2. However, we cannot rely on it directly as now we have to deal with the triangular attention.

Assume for contradiction that there exists a one-layer Transformer $T$ such that (a) it solves the quotient binary relation composition task; (b) it has $n^{o(1)}$ standard-attention heads, $n^{o(1)}$ triangular-attention

heads, $n^{o(1)}$ embedding dimension, and $n^{o(1)}$-size output MLP. Without loss of generality, let $n$ be even and fix $s$ be such that $n = 2s + 2$. Given two binary words $p = p_1 \ldots p_s, q = q_1 \ldots q_s \in \{0, 1\}^s$, we define an instance $(A(p), B(q), \mathsf{col})$ of the quotient binary relation composition task by setting:

$$A_{1,2+j} = p_j, \qquad B_{2+s+j,2} = q_j,$$

for $j \in [s]$, and letting all the other entries of the matrices $A, B$ to be 0. The coloring function is defined by $\mathsf{col}(1) = \mathsf{col}(2) = 1$ and

$$\mathsf{col}(2 + j) = \mathsf{col}(2 + s + j) = 2 + j$$

for $j \in [s]$. An example of this construction for $s = 4$ is given in Figure 3.

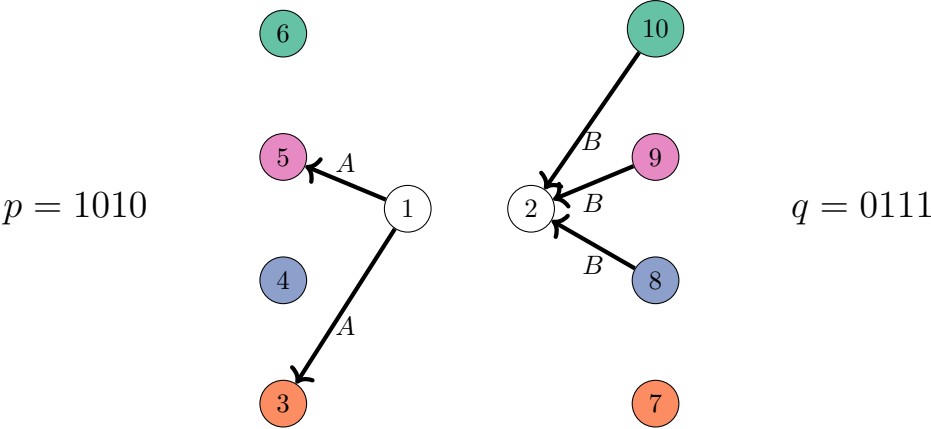

Figure 3: An example of the construction. Nodes apart from 1 and 2 are split into 2 equal groups – 3, 4, 5, 6 and 7, 8, 9, 10. If $A_{ij} = 1$ (resp., $B_{ij} = 1$), we draw an $A$-labeled (resp., a $B$-labeled) edge from $i$ to $j$. The $A$-edges can only go from 1 to 3,4,5,6, and the word $p$ determines, which of these edges are present. Likewise, the $B$-edges only go from 7, 8, 9, 10 to 2, according to whether $q$ has 1 or 0 in the corresponding position. Nodes 3 and 7, 4 and 8, 5 and 9, 6 and 10 have the same color but different pairs have a different color. Therefore, the only way we can have $(B \circ A/c)_{12} = 1$ is when 1 has an $A$-edge to some node on the left, and the node of the same color from the right has a $B$-edge to 2. In the example from the figure, this is true for 5 and 9 (which happens because $p$ and $q$ both have 1 in the third position).

We claim that for the instance $(A(p), B(q), \mathsf{col})$, the value of the quotient binary relation composition at the pair $(1, 2)$ is defined by the equation:

$$(B(q) \circ A(p)/\mathsf{col})_{12} = \mathrm{Disj}(p, q), \qquad (25)$$

where $\mathrm{Disj}(p, q)$ is 1 if and only if there is a position where both $p$ and $q$ have 1. Indeed, if $\mathrm{Disj}(p, q) = 1$, taking $j \in [s]$ such that $p_j = q_j = 1$ and then setting $k_1 = 2 + j, k_2 = 2 + s + j$, we obtain that $A_{1k_1} = p_j = q_j = B_{k_2 2} = 1$ and $\mathsf{col}(k_1) = \mathsf{col}(k_2) = 2 + j$, which implies $(B(q) \circ A(p)/\mathsf{col})_{12} = 1$. On the other hand, if $(B(q) \circ A(p)/\mathsf{col})_{12} = 1$, then for some $k_1 \neq k_2$ we have $A_{1k_1} = B_{k_2 2} = 1$ and $\mathsf{col}(k_1) = \mathsf{col}(k_2)$. Since $A_{1k_1} = 1, B_{k_2 2} = 1$, we have $k_1 = 2 + j_1$ and $k_2 = 2 + s + j_2$ for some $j_1, j_2 \in [s]$. Since $2 + j_1 = \mathsf{col}(k_1) = \mathsf{col}(k_2) = 2 + j_2$, we derive that $j_1 = j_2 = j$. Hence, we get $p_j = A_{1k_1} = B_{k_2 2} = q_j$ and $\mathrm{Disj}(p, q) = 1$, as required.

We now put the instance $(A(p), B(q), \mathsf{col})$ to our Transformer $T$ and look at its output int the token indexed by $(1, 2)$. By (25), we have:

$$\mathsf{sign}(y_{12}) = \mathrm{Disj}(p, q).$$

We now look at how the output $y_{12}$ is computed in our Transformer on such input. The key observation is that, for any $h = 1, \ldots, H$, the output of the $h$-th attention head in position $(1, 2)$ can be written similarly to (20) as a fraction, where some terms depend solely on $p$ and others solely on $q$:

$$a_{12}^{(h)} = \frac{\alpha^{(h)}(p) + \beta^{(h)}(q) + \gamma^{(h)}}{\lambda^{(h)}(p) + \mu^{(h)}(q) + \nu^{(h)}}, \qquad \alpha^{(h)}(p), \beta^{(h)}(q), \gamma^{(h)} \in \mathbb{R}^d, \qquad \lambda^{(h)}(p), \mu^{(h)}(q), \nu^{(h)} \in \mathbb{R}$$

*regardless of whether the $h$-th attention head uses the standard or the triangular attention.* Indeed, for the case of the standard attention, this is by the same computation as in the proof of Theorem 3.2. Now, for the case of the triangular attention, the same computation goes through but for pairs of tokens of the form $(x_{1\ell}, x_{\ell 2})$ instead of individual tokens. It remains to notice that for $\ell = 3, \ldots, s+2$, the value of this pair is determined by $p$, and for $\ell = s+3, \ldots, 2s+2$, the value of this pair is determined by $q$ (and for $s = 1, 2$, the value of this pair is fixed).

The rest of the proof is identical to the corresponding part in the proof of Theorem 3.2. Similarly to (21), we can now write:

$$\text{Disj}(p, q) = \text{sign}\left(\mathcal{N}\left(x_{1,2} + W_O \begin{pmatrix} \frac{\alpha^{(1)}(p)+\beta^{(1)}(q)+\gamma^{(1)}}{\lambda^{(1)}(p)+\mu^{(1)}(q)+\nu^{(1)}} \\ \vdots \\ \frac{\alpha^{(H)}(p)+\beta^{(H)}(q)+\gamma^{(H)}}{\lambda^{(H)}(p)+\mu^{(H)}(q)+\nu^{(H)}} \end{pmatrix}\right)\right). \tag{26}$$

Then we define a hypothesis class $H$ by considering parts of (26) that depend on $p$ as inputs and parts that depend on $q$ as parameters. On the one hand, since $d, H$ and the size of $\mathcal{N}$ are assumed to be $n^{o(1)}$, the VC dimension of this class is $n^{o(1)}$ by Theorem 2.3 in [9]. On the other hand, its VC dimension is lower bounded by the VC dimension of the set of columns of the matrix $\text{Disj}(p, q)$, which is at least $s = n/2 - 1$ as established in the proof of Proposition B.2.

## C    Experimental Setup

### C.1    Datasets

We create dedicated datasets to evaluate our models across all four tasks. Each task consists of $5 \times 10^4$ examples. Below, we detail the data generation process for each task, with explanations of key components and structures.

#### C.1.1    Function Composition

The task of function composition involves determining whether a specific condition holds for a given sequence derived from a function $f$. Each example in the dataset is represented as a tuple $(X, y)$, where:

- $X = (\perp, f(0), f(1), \ldots, f(n-1))$ is an input sequence of length $n + 1$. The first token, $\perp$, is a query token indicating the position where the output is required.
- $n$ is sampled uniformly from the range $[N_{\min}, N_{\max}]$, with $N_{\min} = 25$ and $N_{\max} = 30$.
- $y \in \{0, 1\}$ is a binary label that indicates whether the condition $f(f(0)) = 0$ is satisfied.

The dataset generation process ensures that the sequences are random but incorporates specific constraints to maintain diversity and balance (approximately 50% positive labels). Algorithm 1 outlines the data generation procedure.

---

**Algorithm 1** Dataset Generation for Function Composition

---

**Input:** $N_{\min}, N_{\max}$
**Output:** Dataset
**for** _ = 1 **to** $5 \times 10^4$ **do**
    Sample $n \sim \text{Uniform}(N_{\min}, N_{\max})$
    Generate random sequence $X = x_0 x_1 \ldots x_{n-1}$ with $x_i \sim \text{Uniform}(\{0, 1, 2, \ldots, n-1\})$
    Sample $y \sim \text{Uniform}(\{0, 1\})$
    **if** $y = 1$ **and** $x_{x_0} \neq 0$ **then**
        Set $x_{x_0} \leftarrow 0$ {Ensure $f(f(0)) = 0$}
    **else if** $y = 0$ **and** $x_{x_0} = 0$ **then**
        Set $x_0 \sim \text{Uniform}(i \in \{1, 2, \ldots, n-1\} \mid x_i \neq 0)$ {Ensure $f(f(0)) \neq 0$}
    **end if**
    Prepend query token: $X \leftarrow (\perp, X)$
    Add $(X, y)$ to the dataset
**end for**

---

**Explanation of Examples (Table 2)** In the sequence $(\perp, 3, 0, 5, 1, 0, \ldots)$, the label $y = 0$ implies that the condition $f(f(0)) = 0$ does not hold. Specifically: $f(0) = 3$, and $f(3) = 1 \neq 0$. Thus, the condition $f(f(0)) = 0$ is *false*. In the sequence $(\perp, 4, 1, 3, 5, 0, \ldots)$, the label $y = 1$ indicates that the condition $f(f(0)) = 0$ is satisfied. Specifically: $f(0) = 4$, and $f(4) = 0$. Thus, the condition $f(f(0)) = 0$ is *true*.

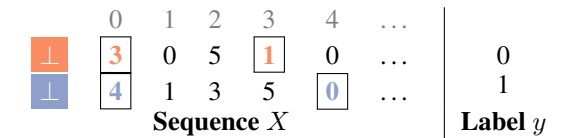

| | 0 | 1 | 2 | 3 | 4 | ... | |
|---|---|---|---|---|---|---|---|
| $\perp$ | 3 | 0 | 5 | 1 | 0 | ... | 0 |
| $\perp$ | 4 | 1 | 3 | 5 | 0 | ... | 1 |
| | | | **Sequence** $X$ | | | | **Label** $y$ |

Table 2: Example sequences for the Function Composition task. The label $y$ depends on whether $f(f(0)) = 0$.

### C.1.2 Binary Relation Composition

The task of binary relation composition involves determining whether a transitive relation exists between two elements in a binary relation matrix. Given two relations $A, B \in \{0, 1\}^{m \times m}$, where $m = \sqrt{n}$, and a pair of elements $i, j \in \{0, 1, ..., m - 1\}$, the goal is to check whether there exists another element $k \in \{0, 1, ..., m - 1\}$ such that both relations $A_{ik}$ and $B_{kj}$ hold. For simplicity, we define $R \in \{0, 1\}^{m \times m}$ and set $A = R$ and $B = R$. Each example in the dataset is represented as a tuple $(X, Y)$, where:

- $X = \mathsf{flatten}(R)$ is an input sequence of length $n = m^2$ of a flatten boolean matrix $R \in \{0, 1\}^{m \times m}$ representing the binary relation. Each element $R_{ij}$ is independently set to 1 with a probability $P$ between 0 to 1, where $P$ is a parameter independent on the input length.

- $m$ is sampled uniformly from the range $[N_{\min}, N_{\max}]$, with $N_{\min} = 6$ and $N_{\max} = 8$.

- $Y \in \{0, 1\}^n$ is a list of binary labels that at position $k = i \times m + j$ indicates whether there is a composition of relations between element $i$ and $j$ is satisfied.

The dataset generation process ensures randomness in $X$ while adhering to the constraints for $Y$. For $m$ in the range $[N_{\min}, N_{\max}]$, the probability $P$ is set to $32.5\%$ to achieve a balanced proportion of positive labels. Algorithm 2 provides the detailed data generation procedure.

---

**Algorithm 2** Dataset Generation for Binary Relation Composition

---

**Input:** $N_{\min}, N_{\max}, P$
**Output:** Dataset
**for** _ $= 1$ **to** $5 \times 10^4$ **do**
    Sample $m \sim \mathrm{Uniform}(N_{\min}, N_{\max})$
    Generate boolean matrix $R \in \{0, 1\}^{m \times m}$, where each element $R_{ij} = 1$ with probability $P$
    Initialize $Y = [\quad]$ {An empty list for labels}
    **for** $i = 0$ **to** $m - 1$ **do**
        **for** $j = 0$ **to** $m - 1$ **do**
            **if** there exists $k$ such that $R_{ik} = 1$ **and** $R_{kj} = 1$ **then**
                Append 1 to $Y$
            **else**
                Append 0 to $Y$
            **end if**
        **end for**
    **end for**
    Add $(\mathsf{flatten}(R), Y)$ to the dataset
**end for**

---

**Explanation of Examples (Table 3)** Consider the binary relation matrix $R$ shown in Table 3: For $i = 0$, $j = 4$, there exists $k = 2$ such that $R_{02} = 1$ and $R_{24} = 1$. Thus, $Y_{0 \times 6 + 4} = 1$. For $i = 5$, $j = 0$, $R_{50} = 1$ is the unique candidate, but $R_{00} = 0$. Thus, $Y_{5 \times 6 + 0} = 0$.

|   | 0 | 1 | 2 | 3 | 4 | 5 |   | 0 | 1 | 2 | 3 | 4 | 5 |
|---|---|---|---|---|---|---|---|---|---|---|---|---|---|
| 0 | 0 | 0 | 1 | 1 | 0 | 0 |   | 1 | 0 | 1 | 0 | 1 | 0 |
| 1 | 0 | 1 | 1 | 0 | 0 | 0 |   | 1 | 1 | 1 | 0 | 1 | 0 |
| 2 | 1 | 0 | 1 | 0 | 1 | 0 |   | 1 | 0 | 1 | 1 | 1 | 0 |
| 3 | 0 | 0 | 0 | 0 | 0 | 0 |   | 0 | 0 | 0 | 0 | 0 | 0 |
| 4 | 0 | 0 | 1 | 1 | 0 | 0 |   | 1 | 0 | 1 | 0 | 1 | 0 |
| 5 | 1 | 0 | 0 | 0 | 0 | 0 |   | 0 | 0 | 1 | 1 | 0 | 0 |
|   |   |   | **Matrix $R$** |   |   |   |   |   |   | $R \circ R$ |   |   |   |

Table 3: Examples of binary relation matrix and corresponding composition. Here, $X = \mathsf{flatten}(R)$ and $Y = \mathsf{flatten}(R \circ R)$.

### C.1.3  Match3

The Match3 task involves determining a target label sequence $Y$ for a given input sequence $X$. Each example consists of an input sequence $X$ of size $n$ and a label sequence $Y$ of the same size, where for every index $i \in \{0, 1, ..., n-1\}$, $Y_i$ represents the target value for $X_i$. Each example is represented as a tuple $(X, Y)$, where:

- $X = (x_0, x_1, \ldots, x_{n-1})$ is a pseudo-random sequence of integers sampled from the range $[0, M-1]$, where $M = 37$.

- $Y = (y_0, y_1, \ldots, y_{n-1})$ is a binary sequence, with $y_i \in \{0, 1\}$, where each value $y_i$ indicates if the token $x_i$ satisfied the Match3 condition in $X$.

- $n$ is sampled uniformly from the range $[N_{\min}, N_{\max}]$, where $N_{\min} = 30$ and $N_{\max} = 35$.

The dataset generation process aims to balance the distribution of ones in $Y$ across four predefined bins corresponding to percentage ranges: $[0, 25)\%$, $[25, 50)\%$, $[50, 75)\%$, and $[75, 100]\%$. Algorithm 3 outlines the data generation procedure.

---

**Algorithm 3** Dataset Generation Algorithm for Match3

**Input:** $N_{\min}, N_{\max}, D$ {Input $D$ is the dataset size}
**Output:** Dataset
Initialize four empty bins {Each bin corresponding to percentage ranges of ones in sequences: $[0, 25)\%$, $[25, 50)\%$, $[50, 75)\%$, and $[75, 100]\%$}
$N_b \leftarrow (D/10)/4$
**for** $i = 1$ **to** $5 \times 10^3$ **do**
    Randomly select skewness $\sim$ Uniform$(1, 40)$ {An initial percentage distribution of ones in the sequence}
    Sample $n \sim$ Uniform$(N_{\min}, N_{\max})$
    Generate a pseudo-random sequence $X = (x_0, x_1, \ldots, x_{n-1})$, where $x_i \sim$ Uniform$(\{0, 1, \ldots, M-1\})$ ensuring the percentage of tokens that satisfied Match3 condition is at least skewness
    Compute $Y = (y_0, y_1, \ldots, y_{n-1})$ based on Match3 condition
    Calculate the percentage of ones in $Y$
    Add $(X, Y)$ to the corresponding bin if size(bin) $< N_b$
**end for**
**for** each bin in bins **do**
    **while** size(bin) $\neq (D/4)$ **do**
        Randomly sample an example $(X, Y)$ from bin
        Apply the same permutation to $X$ and $Y$
        Add the permuted pair $(X^*, Y^*)$ to bin
    **end while**
**end for**
Add all examples from the bins to the dataset

---

**Explanation of Examples (Table 4)**  Consider the example sequence $X$ and its corresponding label $Y$ in Table 4:

- The input sequence $X = (6, 9, 9, 9, 7, 10, 9, 34, 9, 9, 30, \dots)$ contains pseudo-random integers between $0$ and $M - 1$ ($M = 37$).

- The label sequence $Y = (1, 0, 0, 0, 1, 1, 0, 1, 0, 0, 1, \dots)$ has a skewed distribution of ones.

- For instance, $y_5 = 1$ indicates that the element $x_5 = 10$ satisfies the property, because $x_7 = 34$ and $x_{10} = 30$ holds, $x_5 + x_7 + x_{10} = 74 = 2 \times M$.

| | 0 | 1 | 2 | 3 | 4 | 5 | 6 | 7 | 8 | 9 | 10 | ... |
|---|---|---|---|---|---|---|---|---|---|---|---|---|
| **Sequence** $X$ | 6 | 9 | 9 | 9 | 7 | 10 | 9 | 34 | 9 | 9 | 30 | ... |
| **Label** $Y$ | 1 | 0 | 0 | 0 | 1 | 1 | 0 | 1 | 0 | 0 | 1 | ... |

Table 4: Example sequence for Match3 task with $M = 37$.

### C.1.4 Quotient Binary Relation Composition

The task of quotient binary relation composition involves determining whether a transitive relation exists between two elements in a binary relation matrix while incorporating an additional constraint based on a coloring function $\mathsf{col} : [0, m-1] \to [0, m-1]$. For simplicity, we define $R \in \{0, 1\}^{m \times m}$ and set $A = R$ and $B = A^T$. Each example in the dataset is represented as a tuple $(X, Y)$, where:

- $X = \mathsf{flatten}(R)$ is an input sequence of length $n = m^2$ obtained by flattening the boolean matrix $R$, where each element $R_{ij}$ is independently set to 1 with probability $P$, which is independent of the input length.

- $\mathsf{col}$ is a function that assigns a unique color to each element in $[0, m - 1]$, with colors sampled uniformly from the range $[0, m - 1]$.

- $m$ is sampled uniformly from the range $[N_{\min}, N_{\max}]$, with $N_{\min} = 6$ and $N_{\max} = 8$.

- $Y \in \{-100, 0, 1\}^n$ is a list of binary labels, where the position $k = i \times m + j$ indicates whether the quotient binary relation composition between elements $i$ and $j$ is satisfied if and only if $i \neq j$, otherwise $Y_k = -100$.

The dataset generation process ensures randomness in $R$ while adhering to the constraints for $Y$. For $m$ in the range $[N_{\min}, N_{\max}]$, the probability $P$ is set to $43.3\%$ to achieve a balanced proportion of positive labels. Algorithm 4 provides the detailed data generation procedure.

---

**Algorithm 4** Dataset Generation for Quotient Binary Relation Composition

---

**Input:** $N_{\min}, N_{\max}, P$
**Output:** Dataset
**for** _ $= 1$ **to** $5 \times 10^4$ **do**
    Sample $m \sim \mathrm{Uniform}(N_{\min}, N_{\max})$
    Generate boolean matrix $R \in \{0, 1\}^{m \times m}$ where each element $R_{ij} = 1$ with probability $P$
    Generate a coloring function $\mathsf{col} : [0, m - 1] \to [0, m - 1]$ by assigning colors uniformly
    Initialize $Y = [\quad]$ {An empty list for labels}
    **for** each pair $(i, j)$ in $[0, m - 1] \times [0, m - 1]$ **do**
        Append 1 to $Y$ if $i \neq j$ and there exist $k_1, k_2$ such that:
            $R_{ik_1} = 1, R_{jk_2} = 1, \mathsf{col}(k_1) = \mathsf{col}(k_2)$, and $k_1 \neq k_2$
        Otherwise, if $i \neq j$, append 0 to $Y$, else append $-100$ to $Y$
    **end for**
    Add $(\mathsf{flatten}(R), Y)$ to the dataset
**end for**

---

**Explanation of Examples (Table 5)** Consider the binary relation matrix $R$ shown in Table 5. For $i = 2, j = 4$, there exist elements $k_1 = 4$ and $k_2 = 5$ such that $R_{24} = 1$, $R_{45} = 1$, and $\mathsf{col}(4) = 2 = \mathsf{col}(5)$, with $k_1 \neq k_2$. Thus, $Y_{2 \times 7 + 4} = 1$.

| | 0 | 1 | 2 | 3 | 4 | 5 | 6 | | col | 0 | 1 | 2 | 3 | 4 | 5 | 6 |
|---|---|---|---|---|---|---|---|---|---|---|---|---|---|---|---|---|
| 0 | 0 | 1 | 1 | 1 | 1 | 0 | 0 | | 5 | | 1 | 1 | 1 | 1 | 0 | 0 |
| 1 | 0 | 0 | 0 | 0 | 0 | 1 | 1 | | 4 | 0 | | 0 | 0 | 0 | 1 | 1 |
| 2 | 1 | 1 | 0 | 0 | [1] | 1 | 0 | | 5 | 1 | 1 | | 0 | [1] | 1 | 0 |
| 3 | 0 | 0 | 0 | 0 | 0 | 1 | 1 | | 1 | 0 | 0 | 0 | | 0 | 1 | 1 |
| 4 | 1 | 0 | 0 | 0 | 0 | [1] | 1 | | 2 | 1 | 0 | 0 | 0 | | 1 | 1 |
| 5 | 0 | 0 | 1 | 1 | 0 | 0 | 1 | | 2 | 0 | 0 | 1 | 1 | 0 | | 1 |
| 6 | 0 | 1 | 0 | 0 | 0 | 1 | 0 | | 3 | 0 | 1 | 0 | 0 | 0 | 1 | |
| | | | **Matrix** $R$ | | | | | | | | | | $R^T \circ R/\mathsf{col}$ | | | |

Table 5: Examples of binary relation matrix and corresponding quotient composition.

## C.2 Experimental Protocol

**Dataset Splitting.** The dataset is split into a training and validation set. We randomly select 90% of the data for training, and the remaining 10% is used for validation. The validation set is used to monitor the model's performance as well as test set. The validation data is kept within the same distribution as the training set to ensure that the evaluation is conducted in an *in-distribution* manner.

**Evaluation.** The evaluation metric for all tasks is accuracy. To compute the accuracy during training or evaluation, we first calculate the accuracy for each batch individually. This is done by dividing the number of correctly predicted labels by the total number of labels in that batch. These per-batch accuracies are then aggregated across all batches within an epoch. The overall accuracy for the epoch is obtained by taking the mean of the per-batch accuracies. Note that this is not equivalent to just taking the number of correctly predicted labels divided by the number of all labels in the whole dataset (due to variable lengths of examples). Both approaches converge to the same value as the number of examples grows.

**Training Setting.** We fix random seeds across all experiments. Each task and model configuration is evaluated using 8 different random seeds, and the reported results include the median across these runs. This approach mitigates the effects of random weight initialization and stochastic data sampling during training. Additionally, training parameters such as learning rate, batch size, and training duration equally specified for each task across model, as summarized in Table 6.

**Implementation Details** We implement all models using PyTorch framework [1] with Opt-Einsum library [5]. We employ AdamW optimizer for all training tasks without a learning rate scheduler, ensuring a consistent optimization strategy across experiments. We use the Binary Cross-Entropy loss as the objective function. In order to handle padding, we used attention masks, preventing the model from attending to padded positions, and thus ensuring no changes in the model's outputs. For the target sequence, we assign a value of $-100$ to positions corresponding to padded tokens. This ensures that during loss and accuracy calculations, only tokens with values other than $-100$ are considered. We achieve this by applying a `mask`, where predictions and targets are filtered as `pred = pred[mask]` and `target = target[mask]`, respectively.

**Hardware and Resource Utilization** We conduct all experiments on high-performance NVIDIA GPUs. Specifically, we execute on *NVIDIA A100* GPUs with 80GB of memory tasks requiring extensive computational resources, such as Match3 and Quotient Binary Relation Composition. For tasks with lower computational demands, such as Function Composition and Binary Relation Composition, We use *NVIDIA A40* GPUs with 48GB of memory.

# D Further Experiments

We evaluate the computational performance of attention mechanisms exclusively on an *NVIDIA RTX A6000* GPU. This analysis focuses on three metrics: (a) forward pass time, (b) GPU utilization, and (c) memory utilization. These metrics provide insights into the computational efficiency and hardware constraints associated with different configurations of hidden dimension and input length. Below, we detail the evaluation methodology and the observed limitations in the Figure 4.

| Task | Model | $d$ | $h$ | $B$ | $\rho$ | $T$ | $p$ |
|------|-------|-----|-----|-----|--------|-----|-----|
| Function Composition | Standard | 16 | 1 | 2500 | $1 \cdot 10^{-3}$ | 1000 epochs | 0.3 |
| | Third-Order | 16 | 1 | 2500 | $1 \cdot 10^{-3}$ | 1000 epochs | 0.3 |
| | Strassen | 16 | 1 | 2500 | $1 \cdot 10^{-3}$ | 1000 epochs | 0.3 |
| Binary Relation Composition | Standard | 16 | 1 | 2500 | $1 \cdot 10^{-3}$ | 200 epochs | 0.3 |
| | Triangular | 16 | 1 | 2500 | $1 \cdot 10^{-3}$ | 200 epochs | 0.3 |
| | Third-Order | 16 | 1 | 2500 | $1 \cdot 10^{-3}$ | 200 epochs | 0.3 |
| | Strassen | 16 | 1 | 2500 | $1 \cdot 10^{-3}$ | 200 epochs | 0.3 |
| Match3 | Standard | 128 | 2 | 2500 | $1 \cdot 10^{-3}$ | 500 epochs | 0.4 |
| | Third-Order | 128 | 2 | 2500 | $1 \cdot 10^{-3}$ | 500 epochs | 0.4 |
| | Strassen | 128 | 2 | 2500 | $1 \cdot 10^{-3}$ | 500 epochs | 0.4 |
| Quotient Binary Relation Composition | Standard | 16 | 1 | 2000 | $1 \cdot 10^{-3}$ | 3000 epochs | 0.3 |
| | Triangular | 16 | 1 | 2000 | $1 \cdot 10^{-3}$ | 3000 epochs | 0.3 |
| | Third-Order | 16 | 1 | 2000 | $1 \cdot 10^{-3}$ | 3000 epochs | 0.3 |
| | Strassen | 16 | 1 | 2000 | $1 \cdot 10^{-3}$ | 3000 epochs | 0.3 |
| COGS | Standard | 68 | 4 | 100 | $5 \cdot 10^{-4}$ | 200 epochs | 0.1 |
| | Strassen | 64 | 4 | 100 | $5 \cdot 10^{-4}$ | 200 epochs | 0.1 |

Table 6: Training parameter settings for tasks and models. For all models and tasks, we run experiments on 8 different seeds with only one attention layer (except for the COGS dataset where we use 3 layers). Here $d$ is embedding dimension, $h$ is the number of heads, $B$ is the batch size, $\rho$ is the learning rate, $T$ is training duration and $p$ is the dropout outside attention mechanism. We did not use batch normalization for any task.

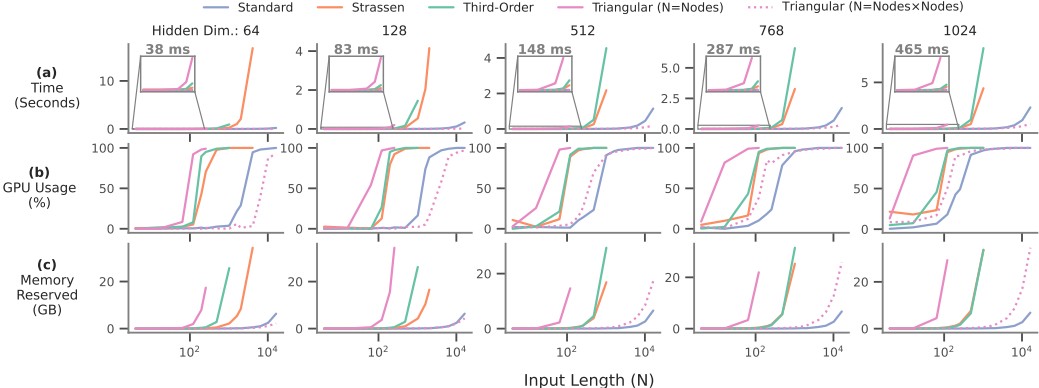

Figure 4: Analysis of computational performance for each attention mechanisms presented in this work: **Standard**, **Strassen**, **Third-Order** and **Triangular** attention. Metrics include (a) forward pass time in seconds, (b) GPU utilization in percentage, and (c) memory reserved in GB on an *NVIDIA RTX A6000*. Experiments are conducted across varying input lengths (4 to 16,384) and five hidden dimensions (64, 128, 256, 768, 1024). Each result represents the average of the median over 8 runs on 100 random sequences (100 tensors with batch size 1) passed through the respective attention mechanisms.

**Time**    We measure the forward pass time as the delta time before and after passing a $1 \times n \times d$ random tensor through the attention mechanism, where $n$ is input length and $d$ hidden dimension. We implement this using `time.time()` function from Python. As we expect, the results indicate that forward pass time increases significantly for higher hidden dimensions and input lengths.

**GPU Usage**    We monitor GPU utilization using the `pynvml` library to query the current CUDA device. Specific configurations of attention mechanisms, such as Strassen attention with large hidden dimensions (e.g., 512 or higher) and long input lengths (e.g., 1600 or higher), result in 100% GPU usage.

**Memory Reserved**    We record the memory reserved during tensor processing using CUDA memory management functions from PyTorch: `torch.cuda.memory_reserved`.

Our computational performance results show that (Figure 4):

- The computational constraints of **Strassen** attention become evident for hidden dimensions of 512 or higher and input lengths exceeding 1600.
- Those of **Third-Order** attention appear with even lower hidden dimensions (64 or 128) and input lengths exceeding 1600.
- **Standard and Triangular** ($n =$Nodes$\times$Nodes) GPU memory usage does not shows a bottleneck even for configurations with hidden dimensions of 2048 and input lengths of 16,384.
- **Triangular** ($n =$Nodes) Limitations appear for hidden dimensions of 1024 or higher and input lengths as low as 196, with increasing severity for longer input sequences.

