# OpenReview forum: "Strassen Attention, Split VC Dimension and Compositionality in Transformers"
_NeurIPS.cc/2025/Conference — NeurIPS 2025 poster_

### Official Review · Reviewer_tYaL · 2025-06-28

**Clarity:** 3
**Significance:** 3
**Originality:** 3
**Rating:** 4
**Confidence:** 4

**Summary:**

This paper studies theoretical limitations of the transformer architecture, and proposes an alternative to attention mechanism - strassen attention. The main contributions are summarized as follows.
1. This work introduces a new technique, called splitting-VC dimension, for transformer size lower bound. This paper proves that any single-layer transformer that computes a function $f$ must have one of embedding dimension, number of attention heads, size of MLP greater than the splitting-VC dimension of $f$, even if the transformer has infinite precision.
2. Furthermore, this paper takes advantage of this new technique to prove lower bounds for reasoning tasks such as function composition, triple-wise correlation, (quotient) binary relation composition. It shows that all three tasks require the model size (max of MLP size, embedding dimension, number of heads) to be at least $n^{\Omega(1)}$ for single-layer transformers.
3. This paper proposes a new alternative to attention mechanism, called strassen attention, that requires $O(n^{\omega})$ time to inference, but overcomes the lower bounds for transformers on the tasks above.
The theoretical results for strassen attention are verified by a series of experiments.

**Questions:**

1. I think it is not hard to see that in terms of representational power, attention < strassen attention < third-order attention? I believe that it is helpful to mention this.
2. Based on question 1, it is interesting that strassen attention outperforms third-order attention in figure 1. Do the authors have some ideas why this would happen?

**Ethical Concerns:**

["NO or VERY MINOR ethics concerns only"]

**Final Justification:**

I think this work is an interesting theory work with some experiments verifying the results. In particular, I find the new lower bound technique pretty interesting. The main limitation, I believe, is the lack of larger scale experiments and that the theory results fail to generalize to multi-layer transformers, which is a bottleneck for many such analysis. Therefore, I suggest a borderline accept.

**Limitations:**

yes

**Quality:**

3

**Strengths And Weaknesses:**

Strengths:
1. Limitations of the transformer architecture, or attention mechanism, is an interesting and important area of research. In addition, this paper provides a concrete and novel technique for transformer lower bounds (compared to previous work that was often based on communication complexity).
2. I think the idea of strassen attention is quite interesting. One can view the standard attention matrix as an $n \times n$ matrix such that each entry corresponds to one pair of key and query $\langle q_i,k_j\rangle$, the third-order attention (defined in [1]) matrix as an $n \times n^2$ matrix such that each entry corresponds to the triple-wise inner product of $q_i,k_{1j},k_{2\ell}$, and here the strassen attention matrix as $\langle q_i,k_{1j}\rangle+\langle q_i,k_{2\ell}\rangle+\langle k_{1j},k_{2\ell}\rangle$. Intuitively, strassen attention is the “middle-ground” of standard attention and third-order attention, where it has strictly stronger representational strength compared to attention but strictly weaker representation strength compared to third-order attention.
3. Theoretical results are verified by experiments. Based on figure 1, it seems like strassen attention outperforms standard attention with respect to accuracy/time, which alleviates the concern of super-quadratic training/inference time.

Weakness:
1. The lower bounds proved are only for single-layer transformers, which weakens the lower bound results. I think a lower bound for multi-layer transformers would be very interesting, although I understand it is a generally difficult problem.
2. Regarding the motivation of splitting-VC dimension (line 174-179), my understanding is that, for tasks like function composition, the input contains both the “real input” and functions and therefore we want to split the input into two parts. Are there more motivations? What is its connection to VC dimension?
3. I think the presentation of section 2 is not clear enough. It could be a personal taste but I think some more explanation (for example, strengths part 2) could be helpful.
4. The tasks/functions here are mostly binary functions and therefore can be parameterized by the splitting-VC dimension. What about other tasks that are not binary? For example next token prediction, translation etc?
5. The experiments are small-scale experiments. It is unclear whether strassen attention based transformers are efficient when the model size gets larger. Are there efficient implementations? The real running time for strassen attention will eventually still based on matrix multiplication on hardware, which takes cubic time still as the $n^{\omega}$ time algorithm exists only in theory.

Overall I find the most interesting parts of this work to be a new technique for proving binary function lower bounds for transformers, and the empirical results which show that strassen attention based transformers outperform vanilla transformers on a series of reasoning tasks. Due to the limitations stated above, I suggest a borderline accept.

[1] Clayton Sanford, Daniel J. Hsu, and Matus Telgarsky. Representational Strengths and Limitations of Transformers. In Alice Oh, Tristan Naumann, Amir Globerson, Kate Saenko, Moritz Hardt, and Sergey Levine, editors, Advances in Neural Information Processing Systems 36.

---

> ### Author Rebuttal · Authors · 2025-07-30
>
> We thank the reviewer for the positive appraisal of our work and greatly appreciate the time invested in reviewing it. Below, we reply to the weaknesses and questions raised by the reviewer.
>
> * _(Weakness 1) The lower bounds proved are only for single-layer transformers, which weakens the lower bound results. I think a lower bound for multi-layer transformers would be very interesting, although I understand it is a generally difficult problem._
>
> The main technical challenge for extending lower bounds to multi-layer softmax encoders is that this requires breakthrough results in computational complexity theory (see Chen, Peng, and Wu, ``Theoretical limitations of multi-layer Transformer’’, Appendix B). The function composition task can be done in 2 constant-size standard-attention layers. For our other 3 tasks we conjecture that they cannot be done with a constant number of layers of n^{o(1)}-size standard attention layers.
>
> * _(Weakness 2) Regarding the motivation of splitting-VC dimension (line 174-179), my understanding is that, for tasks like function composition, the input contains both the “real input” and functions and therefore we want to split the input into two parts. Are there more motivations? What is its connection to VC dimension?_
>
> The main motivation is to avoid communication complexity arguments and considerations of precision. The connection to VC dimension is made precisely by splitting the input of the transformer, which can then be naturally seen as implementing a hypothesis class to which the regular notion of VC dimension applies.  This allows us to invoke results from VC-dimension theory in order to obtain our results.
>
>
> * _(Weakness 4) The tasks/functions here are mostly binary functions and therefore can be parameterized by the splitting-VC dimension. What about other tasks that are not binary? For example next token prediction, translation etc?_
>
> With our method one can obtain lower bounds for any kind of task as long as there is a reduction to a binary function with high splitting-VC dimension. We do so, for instance, for the function composition which is not a binary task. We envision that a similar thing can be done for tasks like next-token prediction.
>
> * _(Question 1) I think it is not hard to see that in terms of representational power, attention < strassen attention < third-order attention? I believe that it is helpful to mention this._
>
> A Strassen-attention layer can be simulated by a third-order attention layer with constant blow-up in the embedding dimension. We do not know whether this can be done in the other direction, but we find it unlikely, given that a Strassen-attention layer can be implemented faster.
>
> * _(Question 2) Based on question 1, it is interesting that strassen attention outperforms third-order attention in figure 1. Do the authors have some ideas why this would happen?_
>
> This is an interesting point. We think this is related to a difference in learnability between both architectures. Strassen attention seems to have a nicer loss landscape for certain tasks, but more work is required in order to confirm this hypothesis.

---

> > ### Comment · Reviewer_tYaL · 2025-08-02
> >
> > I thank the authors for the response. I still think that the new lower bound technique and experimental results are interesting, and the limitations of this work (cannot be extended to multi-layer, need experiments on tasks such as next-token prediction and translation) still exist. I will keep my score as it is.

---

### Official Review · Reviewer_3L74 · 2025-07-01

**Clarity:** 3
**Significance:** 3
**Originality:** 3
**Rating:** 4
**Confidence:** 4

**Summary:**

This paper introduces split-VC dimension as a novel theoretical framework for analyzing transformer expressivity and proposes Strassen attention to address compositional reasoning limitations. The primary contribution establishes rigorous lower bounds proving that one-layer standard transformers cannot solve tasks requiring ternary interactions (function composition, Match3, quotient binary relations) even with infinite precision arithmetic. The split-VC framework adapts classical learning theory to modern architectures, while Strassen attention offers a sub-cubic O(n^2.372) solution through triple-wise token interactions. The theoretical analysis connects complexity measures to architectural constraints through well-structured impossibility proofs.

**Questions:**

1. Can you provide experiments comparing Strassen attention against standard attention with matched parameters (either 5 projection matrices or d'=1.29d)? Include ablations testing V_1=V_2 weight sharing.
2. Will you evaluate on CLUTRR (n>=128) and COGS to demonstrate practical compositional advantages? What performance improvements justify 67% parameter overhead?
3. Please provide GPU runtime measurements (A100/H100) for sequences {1k, 4k, 16k} showing actual crossover points where Strassen outperforms standard attention in wall-clock time.
4. Can you prove lower bounds under realistic assumptions: bounded MLPs (P=O(d^2)) and structured positional encodings (p(i,sigma_i) = q(i) + r(sigma_i) )? How do results change?
5. Are all three interaction terms (f_i g_j + g_j h_k + h_k f_i)  essential? Provide ablations removing individual terms and testing reduced-rank projections.

Score Improvement Criteria: Addressing questions 1-2 with positive results could raise rating to 4 (borderline accept). Demonstrating real-world utility on CLUTRR/COGS with parameter-matched baselines is essential for acceptance.

**Ethical Concerns:**

["NO or VERY MINOR ethics concerns only"]

**Final Justification:**

I have reviewed the author's reply and I decided to increase my score.

**Limitations:**

The authors acknowledge theoretical contributions but inadequately address practical limitations. Missing discussions include:

Scalability beyond synthetic tasks
Numerical stability of Strassen's algorithm in transformer contexts
Energy/memory costs of 67% parameter increase
Applicability to multi-layer architectures (bounds are single-layer only)

Suggested additions: Our theoretical framework analyzes idealized models with unbounded components, limiting direct applicability to practical transformers. The parameter overhead and unvalidated complexity claims require empirical verification before deployment recommendations.

**Paper Formatting Concerns:**

None identified. Paper follows NeurIPS formatting guidelines appropriately.

**Quality:**

2

**Strengths And Weaknesses:**

Strengths

1. The split-VC dimension represents a creative adaptation of classical VC theory to transformer analysis, providing rigorous mathematical foundations for expressivity limitations.
2. Lower bound proofs are technically sound, with elegant reductions (e.g., function composition to Ind_n) demonstrating fundamental limitations of pairwise attention mechanisms.
3. Strassen attention's sub-cubic complexity for ternary interactions shows theoretical promise, correctly identifying that compositional tasks require higher-order token relationships.
4. The framework successfully isolates attention as the computational bottleneck, proving limitations hold even with unbounded MLPs and arbitrary positional encodings.

Weaknesses

1. Strassen attention requires 67% more parameters (5 projection matrices vs. 3), fundamentally confounding architectural innovation with capacity scaling. Without parameter-matched comparisons, performance gains remain unattributable.
2. Exclusive evaluation on synthetic tasks (Match3, quotient relations) provides no evidence of practical utility. The absence of CLUTRR/COGS benchmarks renders compositionality claims speculative.
3. Unbounded MLPs (P >> d,H) and arbitrary positional encodings create a framework analyzing impossible models. Real transformers with bounded components may exhibit entirely different limitations.
4. The O(n^2.372) complexity requires validation under hardware constraints. Memory bandwidth limitations (2.67× accesses), numerical stability, and energy efficiency remain unaddressed.
5. Missing comparisons with modern efficient attention (Performer, BigBird, GQA) and recent compositional reasoning advances (Bhattamishra et al., 2024) undermines positioning within contemporary research.

---

> ### Author Rebuttal · Authors · 2025-07-30
>
> We thank the reviewer for  insightful comments. Below, we reply to the weaknesses and questions raised by the reviewer.
>
> * _(Weakness 1)Strassen attention requires 67% more parameters (5 projection matrices vs. 3), fundamentally confounding architectural innovation with capacity scaling. Without parameter-matched comparisons, performance gains remain unattributable._
>
> We thank the reviewer for pointing out this concern. To address it, we have repeated two experiments from the tasks that require less computation time (Quotient Binary  Relation Composition and Binary Relation Composition). We have compared 1-layer Strassen Attention against 2 layers of Standard Attention, both models with hidden dimension 16. The number of trainable parameters are shown in the following table:
>
> Strassen (1 Layer):  2.5k (Binary Relation Composition), 2.5k (Quotient Binary Relation Composition).
>
> Standard (2 Layer): 3.6k (Binary Relation Composition), 3.6k (Quotient Binary Relation Composition)
>
> The obtained accuracies are presented below:
>
> Strassen (1 layer): 100 % (Binary Relation Composition), 98% (Quotient Binary Relation Composition).
>
> Standard (2 layer): 92 % (Binary Relation Composition), 82% (Quotient Binary Relation Composition).
>
> This analysis revealed that 1-layer Strassen attention still clearly outperforms a 2-layer Standard attention (and about 40% more parameters than the Strassen) in both of the tasks considered. We will include these results in the revised version of the paper.
>
> * _(Weakness 2)Exclusive evaluation on synthetic tasks (Match3, quotient relations) provides no evidence of practical utility. The absence of CLUTRR/COGS benchmarks renders compositionality claims speculative._
>
> We performed an additional experiment on COGS to compare Strassen and Standard attentions under a parameter-matched scenario using the following hyperparameters from [1]: learning rate (5e-4), dropout (0.1), batch size (100), linear warmup scheduler and no early stopping criteria. We have met an accuracy for Standard attention Transformer that was reported by [2] under a similar configuration. All the results are condensed in the following table:
>
> Strassen:  3 (Layers), 64 (hidden dimension), 99k (#Parameters), 72% (Accuracy).
>
> Standard: 3 (Layers), 68 (hidden dimension), 99k (#Parameters), 65% (Accuracy)
>
> This analysis revealed that Strassen attention exhibited a significantly higher performance in the COGS task. We thank the reviewer for suggesting this, and we will include results of this experiment into the revised version of the paper.
>
> [1] Bergen, L, O'Donnell, T., Bahdanau, D.2023. Systematic Generalization with Edge Transformers.
>
> [2] Csordás, R., Irie, K., Schmidhuber, J. 2022. The Devil is in the Detail: Simple Tricks Improve Systematic Generalization of Transformers.
>
> * _(Weakness 3) Unbounded MLPs (P >> d,H) and arbitrary positional encodings create a framework analyzing impossible models. Real transformers with bounded components may exhibit entirely different limitations._
>
> The limitations derived from our lower bounds hold for any kind of MLPs and any kind of positional encoding. This is not a weakness but a strength. In particular they also apply to real transformers. Moreover, our positive results are proved by exhibiting transformers with constant size components, as those used in practice.
>
>
> * _(Question 1) Can you provide experiments comparing Strassen attention against standard attention with matched parameters (either 5 projection matrices or d'=1.29d)? Include ablations testing V_1=V_2 weight sharing._
>
> We thank the reviewer for pointing this concern. We have addressed the comparison with matched parameters in our answer to weakness 1.  We performed the experiment suggested by the reviewer (V_1 = V_2 weight sharing) for the Quotient Binary Relation Composition task.  The accuracy for this experiment was 88%, which is significantly lower than the 98% accuracy obtained without ablation.
>
> * _(Question 2) Will you evaluate on CLUTRR (n>=128) and COGS to demonstrate practical compositional advantages? What performance improvements justify 67% parameter overhead?_
>
> This was addressed in weakness 2.
>
>
> * _(Question 3) Please provide GPU runtime measurements (A100/H100) for sequences {1k, 4k, 16k} showing actual crossover points where Strassen outperforms standard attention in wall-clock time._
>
> Strassen attention enjoys sub-cubic complexity time. It is designed to run faster than other higher-order attention mechanisms which also have a strictly greater expressivity when compared to Standard attention. But none of these more expressive attention mechanisms outperform Standard attention in running time, since Standard attention enjoys quadratic time only. Strassen outperforms the Standard attention not in time but in accuracy.
>
> * _(Question 4) Can you prove lower bounds under realistic assumptions: bounded MLPs (P=O(d^2)) and structured positional encodings (p(i,sigma_i) = q(i) + r(sigma_i) )? How do results change?_
>
> Yes, our lower bounds hold for any kind of MLPs and any kind of positional encoding, including those suggested by the reviewer.  For example, in Theorem 3.7 we show that no one-layer standard attention transformer with n^{o(1)} size can solve the binary relation composition task. Since the transformer suggested by the reviewer is a standard-attention transformer of n^{o(1)} size, it follows that such transformers cannot solve the binary relation composition task.
>
> * _(Question 5) Are all three interaction terms (f_i g_j + g_j h_k + h_k f_i) essential? Provide ablations removing individual terms and testing reduced-rank projections._
>
> From a theoretical point of view, all three interaction terms are essential in our solutions (upper bounds) for the binary relation composition, the match 3, and the quotient binary relation tasks. From an experimental point of view, and following the suggestion of the reviewer, we performed three experiments on the Quotient Binary Relation Composition task, in which we removed the first, the second, and the third interaction term, respectively. In all cases we obtained that accuracy was about 71%, significantly lower than the 98% accuracy obtained without ablation.

---

> > ### Comment · Reviewer_3L74 · 2025-08-07
> >
> > Thank you for your thorough response addressing the parameter matching concerns through controlled experiments comparing 1-layer Strassen against 2-layer Standard attention, and for adding COGS benchmark evaluation. Based on these substantial improvements, particularly the demonstration of practical utility on COGS with parameter-matched baselines meeting the stated criteria, I am raising my score from 3 to 4 (Borderline Accept). The work now presents a meaningful contribution combining novel theoretical insights with empirically validated architectural innovations for compositional reasoning.

---

### Official Review · Reviewer_nDED · 2025-07-02

**Clarity:** 4
**Significance:** 3
**Originality:** 4
**Rating:** 5
**Confidence:** 4

**Summary:**

This paper presents a novel theoretical framework for analyzing the limitations of single-layer Transformers and introduces a new attention mechanism called Strassen Attention that is designed to overcome these limitations. The authors first develop the concept of "Splitting VC dimension" (spvc) to prove that standard one-layer Transformers, even with infinite precision, cannot solve tasks requiring complex compositional reasoning, such as function composition and Match3. Motivated by this, they propose Strassen Attention, a mechanism that efficiently captures interactions between triplets of tokens, resulting in a sub-cubic time complexity - `O(n^ω)` where `ω < 2.372`. The authors provide theoretical proofs demonstrating that a single layer of Strassen Attention can solve these difficult tasks. They further validate their claims through experiments comparing Strassen Attention to standard, triangular, and third-order attention, showing it achieves superior performance and efficiency on a set of challenging synthetic tasks.

**Questions:**

1.  **On the Power of Depth:** Your lower bounds are for one-layer models. Could you elaborate on the technical challenges of extending this analysis to multi-layer Transformers? Do you believe that a constant number of standard attention layers could solve the tasks you've outlined, or would it require a depth polynomial in the sequence length `n`? A brief discussion on this could help frame the importance of mechanisms like Strassen Attention versus simply stacking more layers.

2.  **Numerical Stability and Implementation:** The core of Strassen Attention involves sums of large dot products inside an exponential (`softmax`). Your proof of sub-cubic implementation relies on decomposing this into matrix products. Could you comment on the numerical stability of this approach in practice? For example, in the function composition proof, the dot products are scaled by `n^2`. Did you encounter any issues with vanishing or exploding values during training, and if so, how were they managed? This would be valuable for others seeking to implement your method.

3.  **Generalization to Higher-Order Interactions:** Strassen Attention elegantly handles 3-way interactions. The form `f_i g_j + g_j h_k + h_k f_i` seems inspired by the structure of the tasks. Could this principle be generalized to create an efficient 4-way (or k-way) attention mechanism? For example, would a formulation like `(f_i g_j) + (g_j h_k) + (h_k l_m) + (l_m f_i)` also be implementable with fast matrix multiplication, or does the cyclic structure break down?

**Ethical Concerns:**

["NO or VERY MINOR ethics concerns only"]

**Final Justification:**

The authors have addressed all of my questions and comments. I therefore expect them to revise their manuscript accordingly, including citations of related work, and I maintain my recommendation of Accept (5).

**Limitations:**

Yes. The authors have done a good job of acknowledging the limitations of their work in the "Future directions and limitations" section. They correctly point out that their analysis is based on one-layer models and synthetic tasks, and they identify testing on real-world benchmarks as a crucial next step. They also observe that the learning dynamics and loss landscape of Strassen Attention need further study. The discussion is transparent and appropriately sets the stage for future research.

**Paper Formatting Concerns:**

None.

**Quality:**

4

**Strengths And Weaknesses:**

**Strengths:**

1.  **Original and Important Theoretical Framework:** The introduction of the Splitting VC dimension (`spvc`) is a highly original and powerful method for proving lower bounds. Crucially, it works in the infinite precision regime, which improves previous results that relied on communication complexity and finite precision assumptions. This provides a more fundamental understanding of the architectural bottlenecks of standard attention.

2.  **First Introduction of Strassen Attention:** The proposed Strassen Attention is derived directly from the theoretical limitations identified. The insight to structure the triplet interaction as a sum of pairwise dot-products, which in turn enables a fast implementation via matrix multiplication, is elegant. This makes the mechanism theoretically powerful.

3.  **Comprehensive and Rigorous Evaluation:** The paper excels in its thoroughness. It starts with a new theory, proposes a new architecture, proves its capabilities, and validates everything with carefully designed experiments. The inclusion of the "Quotient Binary Relation Composition" task to theoretically disentangle the capabilities of Strassen Attention from other advanced mechanisms (like triangular attention) adds to the overall storyline of the paper.

4.  **Clarity of Presentation:** Despite the high technical density, the paper is exceptionally well-written. The authors effectively use figures, formal definitions, and step-by-step proof sketches to make complex ideas clear and accessible. The motivation and narrative flow logically from problem to solution.

**Weaknesses:**

1.  **Limited Scope of Lower Bounds (One-Layer):** The primary theoretical limitation is that the lower bounds are only formally proven for single-layer Transformers. While the authors' intuition that these limitations likely extend to multi-layer models is plausible, this remains a conjecture. The possibility that depth could compensate for the lack of breadth in a single layer (i.e., multiple layers of pairwise attention simulating a triplet attention) is a key open question not addressed.

2.  **Reliance on Synthetic Tasks:** The evaluation is conducted exclusively on synthetic "toy" tasks. While these tasks are perfectly designed to isolate and test the specific compositional abilities under investigation, the performance of Strassen Attention on real-world, noisy, and high-dimensional data (e.g., in NLP or vision) is not demonstrated. This leaves a gap between the demonstrated potential and proven real-world utility.

3.  **Expressivity vs. Learnability:** The upper-bound proofs (showing Strassen Attention can solve the tasks) are constructive, demonstrating expressivity. However, they do not guarantee learnability via gradient-based optimization from a random initialization. While the experiments suggest the models are indeed learnable on these tasks, the properties of the loss landscape for more complex problems remain unexplored.

4. **Cite more related works:** The authors should cite related works that study limitations of compositional tasks, like: "Limits of Deep Learning: Sequence Modeling through the Lens of Complexity Theory" by Nikola Zubić, Federico Soldá, Aurelio Sulser, Davide Scaramuzza.

---

> ### Author Rebuttal · Authors · 2025-07-30
>
> We thank the reviewer for the positive appraisal of our work and greatly appreciate the time invested in reviewing it. Below, we reply to the weaknesses and questions raised by the reviewer.
>
>
>
> * _(Weakness 2)Reliance on Synthetic Tasks: The evaluation is conducted exclusively on synthetic "toy" tasks. While these tasks are perfectly designed to isolate and test the specific compositional abilities under investigation, the performance of Strassen Attention on real-world, noisy, and high-dimensional data (e.g., in NLP or vision) is not demonstrated. This leaves a gap between the demonstrated potential and proven real-world utility._
>
> We have performed an additional experiment on COGS, a NLP task that tests compositional abilities. We compared Strassen and Standard attentions under a parameter-matched scenario using the following hyperparameters from [1]: learning rate (5e-4), dropout (0.1), batch size (100), linear warmup scheduler and no early stopping criteria. We have met an accuracy for Standard attention Transformer that was reported by [2] under a similar configuration. All the results are condensed in the following table:
>
> Strassen:  3 (Layers), 64 (hidden dimension), 99k (#Parameters), 72% (Accuracy).
>
> Standard: 3 (Layers), 68 (hidden dimension), 99k (#Parameters), 65% (Accuracy).
>
> This analysis revealed that Strassen attention exhibited a significantly higher performance in COGS task. We thank the reviewer for suggesting this, and we will include results of this experiment into the revised version of the paper.
>
> [1] Bergen, L, O'Donnell, T., Bahdanau, D.2023 .Systematic Generalization with Edge Transformers.
>
> [2] Csordás, R., Irie, K., Schmidhuber, J. 2022. The Devil is in the Detail: Simple Tricks Improve Systematic Generalization of Transformers.
>
>
> * _(Question 1) Your lower bounds are for one-layer models. Could you elaborate on the technical challenges of extending this analysis to multi-layer Transformers? Do you believe that a constant number of standard attention layers could solve the tasks you've outlined, or would it require a depth polynomial in the sequence length n? A brief discussion on this could help frame the importance of mechanisms like Strassen Attention versus simply stacking more layers._
>
>  The main technical challenge for extending lower bounds to multi-layer softmax encoders is that this requires breakthrough results in computational complexity theory (see Chen, Peng, and Wu, ``Theoretical limitations of multi-layer Transformer’’, Appendix B). The function composition task can be done in 2 constant-size standard-attention layers. For our other 3 tasks we conjecture that they cannot be done with a constant number of layers of n^{o(1)}-size standard attention layers.
>
> * _(Question 2) Numerical Stability and Implementation: The core of Strassen Attention involves sums of large dot products inside an exponential (softmax). Your proof of sub-cubic implementation relies on decomposing this into matrix products. Could you comment on the numerical stability of this approach in practice? For example, in the function composition proof, the dot products are scaled by n^2. Did you encounter any issues with vanishing or exploding values during training, and if so, how were they managed? This would be valuable for others seeking to implement your method._
>
>  In the training process we have seen that higher-order models can suffer from exploding values  during optimization, which we avoided by applying a shifting operation to the logits. Also, in order to improve convergence, we have adopted a Kaiming-He distribution for initialization of attention weights. Additionally, we chose to use a pre-layer normalization strategy, which resulted in an improved convergence time for all models. For more details, we refer to the repository link in Section 6 of the paper.
>
> * _(Question 3) Generalization to Higher-Order Interactions: Strassen Attention elegantly handles 3-way interactions. The form f_i g_j + g_j h_k + h_k f_i seems inspired by the structure of the tasks. Could this principle be generalized to create an efficient 4-way (or k-way) attention mechanism? For example, would a formulation like (f_i g_j) + (g_j h_k) + (h_k l_m) + (l_m f_i) also be implementable with fast matrix multiplication, or does the cyclic structure break down?_
>
>  Exactly, the formulation you are suggesting boils down to the product of 4 matrices instead of 3. This is still sub-cubic time. We are grateful for your remark, it opens an interesting direction for future work.

---

> > ### Comment · Reviewer_nDED · 2025-08-06
> > **Accept (5) rating stays**
> >
> > The authors have addressed all of my questions and comments. I therefore expect them to revise their manuscript accordingly, including citations of related work, and I maintain my recommendation of Accept (5).

---

### Official Review · Reviewer_4HHd · 2025-07-03

**Clarity:** 3
**Significance:** 3
**Originality:** 4
**Rating:** 5
**Confidence:** 4

**Summary:**

The paper introduces Strassen Attention, a novel higher-order attention architecture that replaces softmax-attention with a three-way normalized product between exponentiated matrices that can be computed efficiently with Strassen matrix multiplication. The model is presented as an alternative to other “triplet” architectures, like third-order attention and triangle attention, with computational advantages and empirical capabilities. The model is motivated by its ability to solve (theoretically and empirically) tasks like Match3, which are known to be hard for standard transformers. They strengthen this separation by introducing a novel theoretical framework of split-VC dimension for proving the limitations of single-layer transformers with arbitrarily precision to solve these tasks.

**Questions:**

Can you clarify the relationship to [1]? There appears to be substantial overlap in methodology.

Can Strassen and third-order transformers be compared theoretically? i.e. can one be simulated by the other?

Were there any particularly useful empirical lessons learned from training higher-order models? (e.g. differences in learning rate, optimization behavior, etc)

[1] https://arxiv.org/abs/2412.20195?

**Ethical Concerns:**

["NO or VERY MINOR ethics concerns only"]

**Final Justification:**

My positive impression of the paper remains after discussion. My score will stay as is.

**Limitations:**

yes

**Quality:**

4

**Strengths And Weaknesses:**

# Strengths

The Strassen attention architecture is novel, creative, and well-motivated by failures of standard models on synthetic tasks. Figure 2 is particularly helpful for understanding the architecture. The model performs in simple empirical settings. and has strong theoretical motivation.

The split-VC techniques are new and resolve open theoretical questions about the capabilities of arbitrary precision single-layer transformers. (Previous results relied on either burdensome architectural assumptions or exploited bounded bit-precision.)

The experimental sweep includes a wide range of well-known synthetic tasks and demonstrates clear trade-offs between architectures.

# Weaknesses

The computational advantages of Strassen attention are found in the connection to efficient metric multiplication, allowing sub-cubic time performance. However, given the suitability of GPUs and TPUs to multiplying large matrices, can these performance gains be realized with large models on modern hardware?

The empirical results are somewhat limited by their focus on purely synthetic tasks. Did the authors consider experiments on “intrinsically three-wise” practical tasks, perhaps on graph-based or molecular inference?

The theoretical results are limited by the single-layer assumption. Is it likely that the split-VC dimension approach can be extended to deeper models?

## Minor points:
- L276: “runnin-time”
- L1027:  undefined table reference
- Figure 2 is very helpful for understanding the architecture. Can it be moved to the paper body?

---

> ### Author Rebuttal · Authors · 2025-07-30
>
> We thank the reviewer for the positive appraisal of our work and greatly appreciate the time invested in reviewing it. Below, we reply to the weaknesses and questions raised by the reviewer.
>
>
> * _(Weakness 2)* The empirical results are somewhat limited by their focus on purely synthetic tasks. Did the authors consider experiments on “intrinsically three-wise” practical tasks, perhaps on graph-based or molecular inference?_
>
> We have performed an additional experiment on the COGS, a NLP task that tests compositional abilities. We compared Strassen and Standard attentions under a parameter-matched scenario using the following hyperparameters from [1]: learning rate (5e-4), dropout (0.1), batch size (100), linear warmup scheduler and no early stopping criteria. We have met an accuracy for Standard attention Transformer that was reported by [2] under a similar configuration. All the results are condensed in the following table:
>
> Strassen:  3 (Layers), 64 (hidden dimension), 99k (#Parameters), 72% (Accuracy).
>
> Standard: 3 (Layers), 68 (hidden dimension), 99k (#Parameters), 65% (Accuracy).
>
> This analysis revealed that Strassen attention exhibited a significantly higher performance in COGS task. We thank the reviewer for suggesting this, and we will include results of this experiment into the revised version of the paper.
>
> [1] Bergen, L, O'Donnell, T., Bahdanau, D.2023 .Systematic Generalization with Edge Transformers.
>
> [2] Csordás, R., Irie, K., Schmidhuber, J. 2022. The Devil is in the Detail: Simple Tricks Improve Systematic Generalization of Transformers.
>
> * _(Weakness 3) The theoretical results are limited by the single-layer assumption. Is it likely that the split-VC dimension approach can be extended to deeper models?_
>
> The main technical challenge for extending theoretical limitations to multi-layer softmax encoders is that this requires breakthrough results in computational complexity theory (see Chen, Peng, and Wu, ``Theoretical limitations of multi-layer Transformer’’, Appendix B).
>
> * _(Question 2) Can Strassen and third-order transformers be compared theoretically? i.e. can one be simulated by the other?_
>
> A Strassen-attention layer can be simulated by a third-order attention layer with constant blow-up in the embedding dimension. We do not know whether this can be done in the other direction, but we find it unlikely, given that a Strassen-attention layer can be implemented faster.
>
> *  _(Question 3)Were there any particularly useful empirical lessons learned from training higher-order models? (e.g. differences in learning rate, optimization behavior, etc)_
>
> In the training process we have seen that higher-order models can suffer from exploding values  during optimization. We deal with this by applying a shifting operation to the logits. Moreover, in order to improve convergence, we have adopted a Kaiming-He distribution for initialization of attention weights. Additionally, we chose to use a pre-layer normalization strategy, which helped to improve convergence time for all models. For more details, we refer to the repository link in Section 6 of the paper.

---

> > ### Comment · Reviewer_4HHd · 2025-08-05
> >
> > Thank you for your detailed response. I will maintain my score.

---

### Decision · Program_Chairs · 2025-09-17

**Decision:**

Accept (poster)

**Comment:**

The paper makes a solid theoretical contribution (infinite-precision one-layer lower bounds via spVC), introduces a well-motivated attention mechanism with proven capability on targeted tasks, and presents supportive experiments including a new compositional generalization result on COGS with parameter control. However, open questions about multi-layer extensions, practical runtime/scale, and explicit positioning relative to VC-dimension work remain. The camera-ready should (i) clearly distinguish from Kozachinskiy (arXiv:2412.20195), adjusting novelty claims, (ii) expand related work, and (iii) improve clarity/presentation and, if possible, add preliminary runtime/memory measurements.